# Deep learning-enabled segmentation of ambiguous bioimages with deepflash2

Matthias Griebel [1] ✉, Dennis Segebarth [2], Nikolai Stein [1], Nina Schukraft[2], Philip Tovote [2,3], Robert Blum [4] & Christoph M. Flath [1] ✉

Bioimages frequently exhibit low signal-to-noise ratios due to experimental conditions, specimen characteristics, and imaging trade-offs. Reliable segmentation of such ambiguous images is difficult and laborious. Here we introduce deepflash2, a deep learning-enabled segmentation tool for bioimage analysis. The tool addresses typical challenges that may arise during the training, evaluation, and application of deep learning models on ambiguous data. The tool's training and evaluation pipeline uses multiple expert annotations and deep model ensembles to achieve accurate results. The application pipeline supports various use-cases for expert annotations and includes a quality assurance mechanism in the form of uncertainty measures. Benchmarked against other tools, deepflash2 offers both high predictive accuracy and efficient computational resource usage. The tool is built upon established deep learning libraries and enables sharing of trained model ensembles with the research community. deepflash2 aims to simplify the integration of deep learning into bioimage analysis projects while improving accuracy and reliability.

Partitioning images into meaningful segments (e.g., cells, cellular compartments, or other anatomical structures) is one of the most ubiquitous tasks in bioimage analysis[1]. Segmentation facilitates downstream tasks such as detection (both 2D and 3D), tracking, quantification, and statistical evaluation of image features. Depending on the biological analysis setting, we distinguish between semantic and instance segmentation. Semantic segmentation means subdividing the image into meaningful categories[2]. Instance segmentation further differentiates between multiple instances of the same category by assigning the segmented structures to unique entities (e.g., cell 1, cell 2, …). Performing image feature segmentation manually is tedious and time-consuming, which severely limits scalability. Conversely, its automated segmentation promises additional insights, more precise analyses, and more rigorous statistics[2].

Deep learning (DL) has proven to be a flexible method to analyze large amounts of bioimage data[3], and numerous solutions for automated segmentation have been proposed[2,4–10]. Depending on annotated training data, these tools and analysis pipelines are well suited for settings where the observable phenomena exhibit a high signal-to-noise ratio (SNR), for instance, in monodispersed cell cultures. However, the SNR in bioimages is often low, influenced by experimental conditions, sample characteristics, and imaging trade-offs. Such image material is inherently ambiguous, which hampers a reliable analysis. A case in point is the analysis of fluorescent images of complex brain tissue—a core technique in modern neuroscience—which is frequently subject to various sources of ambiguity, such as cellular and structural diversity, heterogeneous staining conditions, and challenging image acquisition processes.

Establishing DL-based segmentation pipelines in low SNR settings means overcoming substantial challenges during model training and evaluation and during the application of the model for the analysis of new images. Training and evaluation challenges commence with the manual annotation process. Here, human experts rely on heuristic criteria (e.g., morphology, size, signal intensity) to cope with low SNRs.

[1]Department of Business and Economics, University of Würzburg, Würzburg, Germany. [2]Institute of Clinical Neurobiology, University Hospital Würzburg, Würzburg, Germany. [3]Center for Mental Health, University Hospital Würzburg, Würzburg, Germany. [4]Department of Neurology, University Hospital Würzburg, Würzburg, Germany. ✉e-mail: matthias.griebel@uni-wuerzburg.de; christoph.flath@uni-wuerzburg.de

Relying on a single human expert's annotations for training can result in biased DL models[11]. At the same time, inter-expert agreement suffers in such settings, which, in turn, leads to ambiguous training annotations[2,12]. Without reliable annotations, there is no stable ground truth, which complicates both model training and evaluation. The application challenge emerges when DL models are deployed for analyzing large numbers of bioimages. This scaling-up step is a crucial leap of faith for users as it effectively means delegating control over the study to a black box system. DL models will generate segmentations for any image. However, the segmentation quality is unknown as the reliability of model generalizations beyond the training data cannot be guaranteed. Selecting a representative subset of images for training and evaluation in a single experiment is already challenging. Maintaining a representative training set across multiple experiments with possibly varying conditions compounds these problems and may eventually prevent reliable automation. For this reason, a viable deployment needs effective quality assurance, or as Ribeiro et al.[13,p. 1135] put it, "if the users do not trust […] a prediction, they will not use it."

In this work, we introduce deepflash2, a DL-based analysis tool that addresses the key challenges for DL-based bioimage analysis. We illustrate the capabilities of deepflash2 using five representative fluorescence microscopy datasets of mouse brain tissue with varying degrees of ambiguity. In addition, we demonstrate the tool's performance on three recent challenge datasets for prostate cancer grading, multi-organ nuclei segmentation, and colonic nuclear instance segmentation and classification. We benchmark the tool against other common analysis tools, achieving competitive predictive performance under the economical usage of computational resources.

## Results

In bioimage analysis, supervised DL models are typically embedded in two consecutive pipelines[2]—training and application. deepflash2 extends these pipelines to better cope with ambiguous data (Fig. 1).

The training and evaluation pipeline serves to fit a model on a given data set. It comprises data annotation, model training, and model validation. In deepflash2, this pipeline integrates annotations from multiple experts and relies on model ensembles to ensure highly accurate and reliable results. The evaluation of the model ensembles is achieved through a two-step evaluation process. The application pipeline leverages a trained DL model to predict the annotations of new images. By facilitating quality monitoring and out-of-distribution detection of new data, deepflash2 goes a step beyond mere prediction.

### Training and evaluation of DL model ensembles

Training builds upon a representative sample of the bioimage dataset under analysis, annotated by multiple experts (the annotations can be performed with any tool). To derive objective training annotations from multi-annotator data, deepflash2 estimates the ground truth (GT) via majority voting or simultaneous truth and performance level estimation (STAPLE[14]). deepflash2 computes similarity scores between expert segmentations and the estimated GT (Dice score for semantic segmentation, average precision for instance segmentation; Section "Evaluation metrics"). These measures of inter-expert variation serve as a proxy for data ambiguity, as shown in the second row of Fig. 2. Well-defined fluorescent labels are typically unanimously annotated (green), whereas ambiguous signals are marked by fewer experts (blue). This causes a high inter-rater variability when different experts annotate the same images[11].

DL model training in deepflash2 capitalizes on model ensembles to ensure high accuracy and reproducibility in the light of data ambiguity[11]. In contrast to recent work on the segmentation of ambiguous data, which focuses on explicitly modeling disagreements among experts[15,16], our training on the estimated GT aims to provide the most objective basis possible for bioimage analysis. Furthermore, the usage of model ensembles facilitates reliable uncertainty quantification[17]. To ensure training efficiency, deepflash2 leverages pretrained feature extractors (encoders) and advanced training strategies (see "Methods", Section "Training Procedure").

The model ensemble predicts semantic segmentation maps, which are evaluated on a hold-out test set (Fig. 2, third row). For instance segmentation tasks, we leverage the *cellpose* library[9], a generalist algorithm for cell and nucleus segmentation. By combining the semantic segmentation maps with *cellpose*'s flow representations, deepflash2 ensures reliable separation of touching objects. In doing so, we extend the original *cellpose* implementation to multichannel input images and multiclass instance segmentation tasks.

Each segmentation is accompanied by a predictive uncertainty map which is summarized by means of the average foreground uncertainty score $U$ (Fig. 2, fourth row; Section "Uncertainty quantification"). These uncertainties are used for quality assurance during application (Section "Application and quality assurance"). To assess the model validity for bioimage analysis, deepflash2 implements the following two-step evaluation process:

1. Absolute performance: Calculating the similarity scores between the predicted segmentations and the estimated GT on the test set. The scores can be accessed via the GUI or Excel/CSV export functions.
2. Relative performance: Relating the performance scores to data ambiguity. The performance scores of individual experts are used to establish the desired performance range and can also be accessed through the GUI or Excel/CSV export.

The proposed evaluation procedure can generally be performed with any analysis tool as long as the required predictive performance is achieved. With regard to the practical application of a DL tool, however, we evaluate the tool's performance along four dimensions: absolute predictive performance as indicated by the similarity to the estimated GT, relative predictive performance compared to the expert annotations, reproducibility of the experiments, and training duration (Fig. 3).

We benchmark the predictive performance of deepflash2 against a select group of well-established algorithms and tools. We utilize

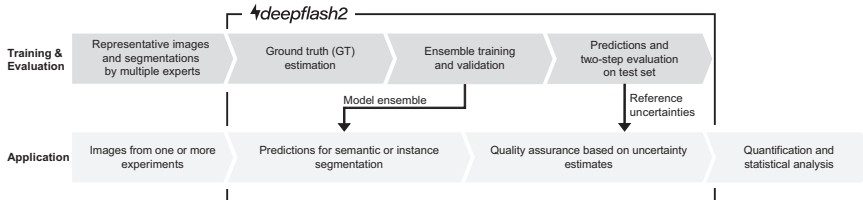

**Fig. 1 | deepflash2 pipelines.** Proposed integration of deepflash2 into the bioimage analysis workflow. In contrast to traditional DL pipelines, deepflash2 integrates annotations from multiple experts and relies on model ensembles for training and evaluation. Additionally, the application pipeline facilitates quality monitoring and out-of-distribution detection for predictions on new data.

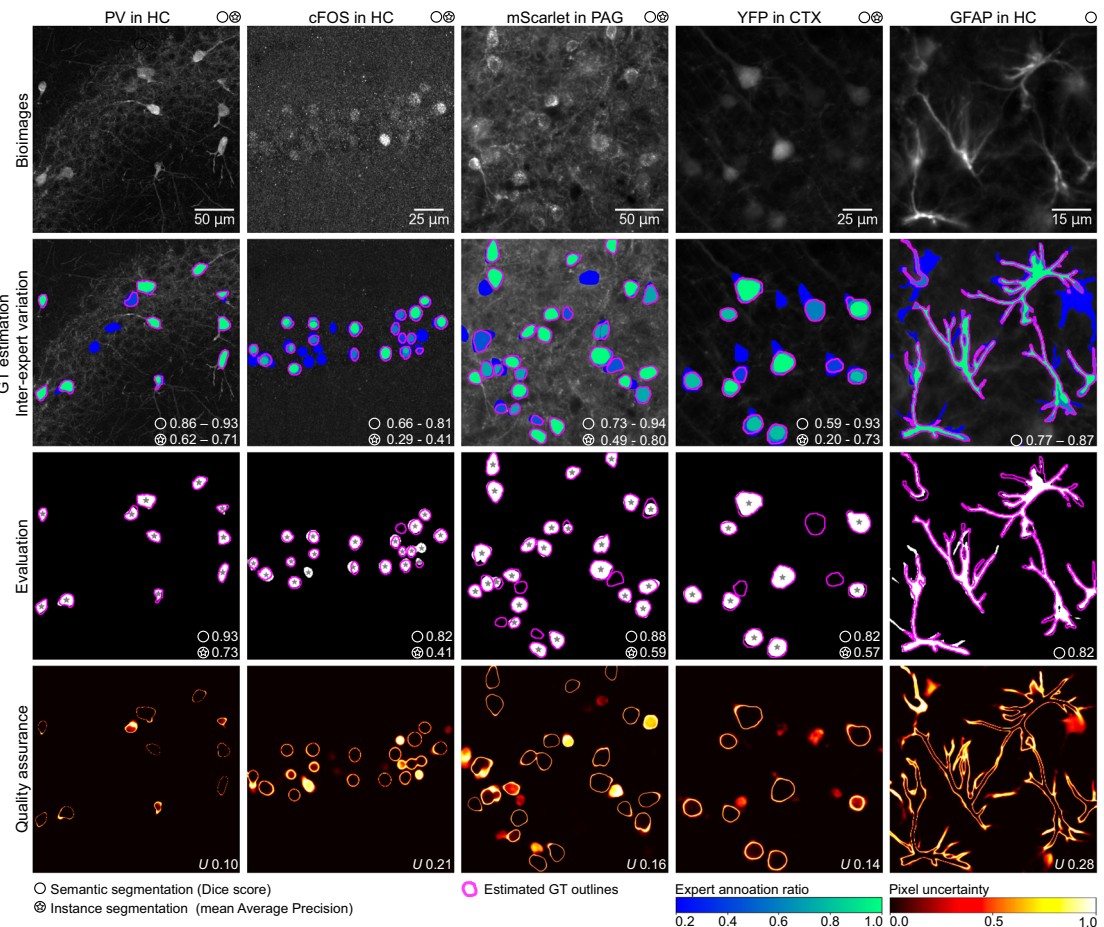

**Fig. 2 | Exemplary results on different immunofluorescence images.** Representative image sections from the test sets of five immunofluorescence imaging datasets (first row) with corresponding expert annotations and ground truth (GT) estimation (second row). The inter-expert variation is indicated with ranges (lowest and highest expert similarity to the estimated GT) of the Dice score (DS) for semantic segmentation and mean Average Precision (mAP) for instance segmentation. The predicted segmentations and the similarity to the estimated GT are depicted in the third row, and the corresponding uncertainty maps and uncertainty scores $U$ for quality assurance are in the fourth row. Areas with a low expert agreement (blue) or differences between the predicted segmentation and the estimated GT typically exhibit high uncertainties. deepflash2 also provides instance (e.g., somata or nuclei)-based uncertainty measures that are not depicted here. The maximum pixel uncertainty has a theoretical limit of 1.

Otsu's method[18] as a simple baseline for semantic segmentation and *cellpose*[9] as a generic baseline for (cell) instance segmentation. Additionally, we consider U-Net[2], *nnunet*[8], and fine-tuned *cellpose* model ensembles. *cellpose* has previously proven to outperform other well-known methods for instance segmentation such as Mask-RCNN[19] or StarDist[20]. For greater clarity, Fig. 3 omits the two baseline models which offered subpar performance (an extensive comparison of all tools is provided in Supplementary Information 2.2).

Across all evaluation datasets, deepflash2 achieves competitive predictive performance for both semantic and instance segmentation tasks. To disentangle the difficulty of the prediction task (driven by data ambiguity) from the predictive performance, we scrutinize the absolute performance by relating it to the underlying expert annotation scores (relative performance). Notably, only deepflash2 achieves human expert performance across all evaluation tasks and, in some cases, even outperforms the best available expert annotation (Fig. 3a, b).

Moreover, Fig. 3c shows that the ensemble-based methods *nnunet* and deepflash2 yield very stable results (high similarity scores between the predicted segmentations of different training runs with different training-validation splits) across all datasets. The U-Net[2], based on a single model, is subject to higher performance variability. The *cellpose* model ensembles exhibit a high variability for the semantic-segmentation-only *GFAP in HC* dataset but yield competitive results on the other (instance segmentation) datasets.

Relying on generic pretrained encoders, deepflash2 model ensembles are trained in less than an hour on machines with state-of-the-art GPUs (free and paid), similar to the pretrained *cellpose* model ensembles (Fig. 3d). Due to dynamic architecture reconfiguration, *nnunet* ensembles cannot leverage pretraining, and training from scratch can last longer than a week.

## Application and quality assurance

During application, scientists typically aim to analyze a large number of bioimages without ground truth information. To establish trust in its predictions, deepflash2 enables quality assurance on image as well as on instance/region level: For quality assurance on image level, the predicted segmentations are sorted by decreasing uncertainty score $U$.

We find that $U$ is a strong predictor for the obtained predictive performance as measured by the Dice score (Fig. 4a). Consequently, $U$ can be used as a proxy for the expected performance on unlabeled data, and the $U$ values of the test set can serve as a reference for the quality assurance procedure (see Section "Quality Assurance" for further details). Note that the model ensembles are solely trained on the estimated GT, that is, there is no longer a concept of ambiguous annotations. However, Fig. 4b confirms that the uncertainty maps capture expert disagreement: Low pixel uncertainty is indicative of high expert agreement, whereas high pixel uncertainty arises in settings where experts submitted ambiguous annotations.

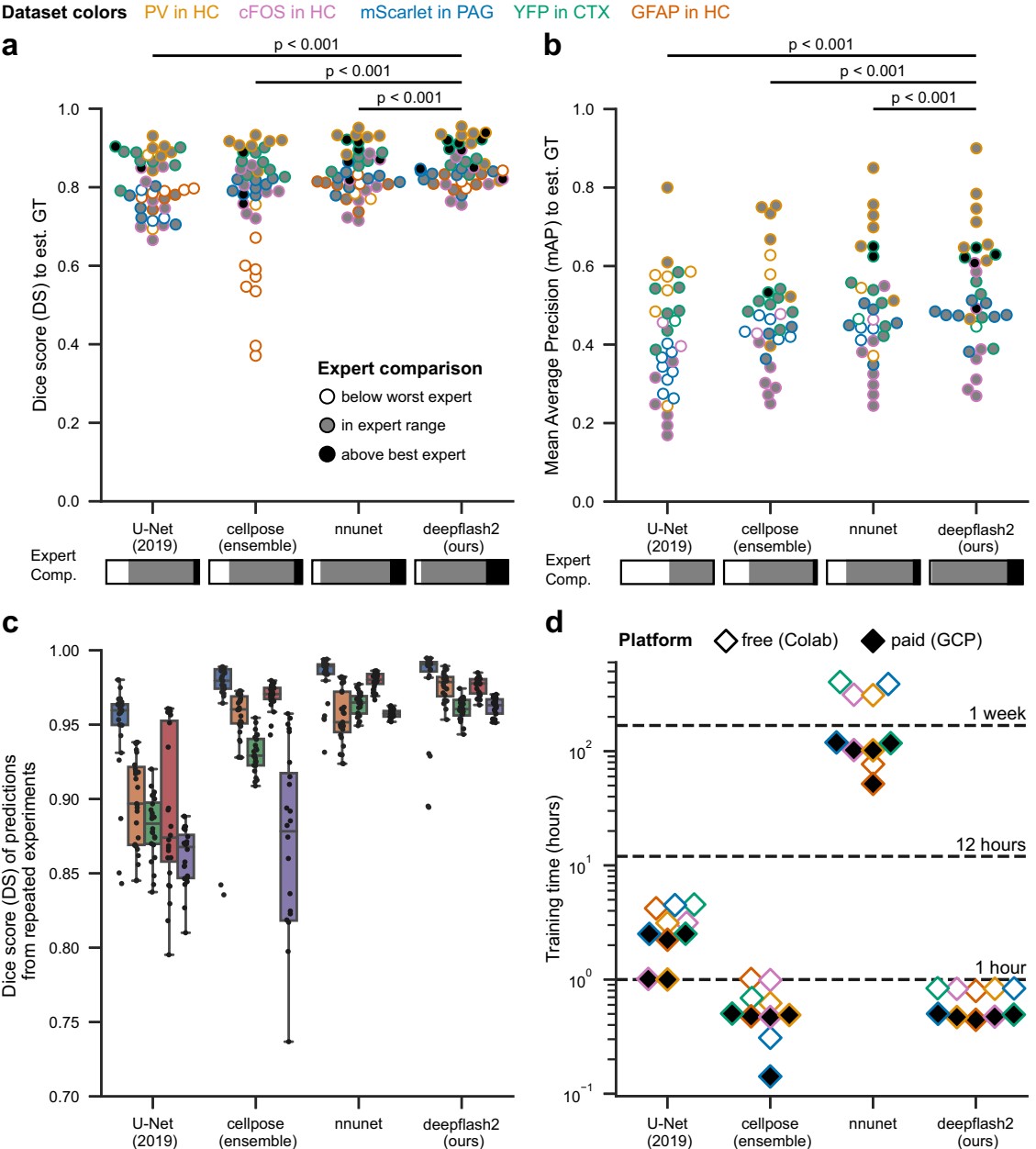

**Fig. 3 | Evaluation of predictive performance, relative performance, reliability, and speed on different immunofluorescence datasets. a, b** Predictive performance on the test sets for **a** semantic segmentation (N = 40, 8 images for each dataset) and **b** instance segmentation (N = 32, 8 images for each depicted dataset except *GFAP in HC*), measured by similarity to the estimated GT. The grayscale filling depicts the comparison against the expert annotation scores. The *p*-values result from a two-sided Wilcoxon signed-rank test (semantic segmentation: p = 0.000170298 for *nnunet*, p = 0.000001405 for *cellpose*, p = 0.000000001 for U-Net (2019); instance segmentation: p = 0.000090546 for *nnunet*, p = 0.000557802 for *cellpose*, p = 0.000000012 for U-Net (2019)). The expert comparison bars below the method names indicate the share of test instances that

scored below the worst expert (white), in expert range (gray), or above the best expert (black). **c** Similarity of the predicted test segmentation masks for three repeated training runs with different training-validation splits (N = 40, 8 images for each dataset). Box plots are defined as follows: the box extends from the first quartile (lower bound of the box) to the third quartile (upper bound of the box) of the data, with a center line at the median. The whiskers extend from the box by at most 1.5x the interquartile range and are drawn down to the lowest and up to the highest data point that falls within this distance. **d** Training speed (duration) on different platforms: Google Colaboratory (Colab, gratuitous Nvidia Tesla T4 GPU) and Google Cloud Platform (GPC, costly Nvidia A100 GPU). Source data are provided as a Source Data file.

In situations with high uncertainty scores, scientists may want to check predictions through manual inspection using the provided uncertainty maps. For semantic segmentation, the uncertainty maps facilitate rapid visual identification of regions where the predicted segmentations are subject to high uncertainties. For instance segmentation tasks, deepflash2 additionally calculates an average uncertainty score for each instance. Subsequently, it allows a single click export-import to ImageJ/Fiji ROIs (regions of interest), with ROIs sorted by their average uncertainty score. This enables a focused

inspection and adjustment of specific instances that are supposedly segmented poorly. Thus, the quality assurance process helps the user prioritize the review of both images and single instances within images that exhibit high uncertainties.

The quality assurance procedure also facilitates the detection of out-of-distribution images, i.e., images that differ from the training data and are thus prone to erroneous predictions. We showcase the out-of-distribution detection on a large bioimage dataset comprising 256 in-distribution images (same properties as training images), 24

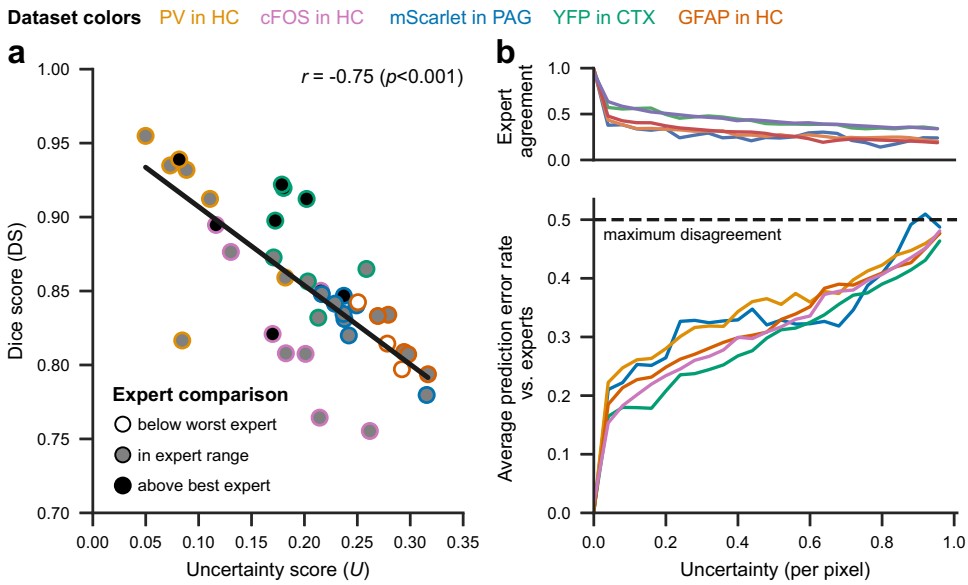

**Fig. 4 | Relationship between expert annotations, uncertainty, and similarity scores. a** Correlation between Dice scores and uncertainties on the test set. We quantify the linear correlation using Pearson's *r* and a two-tailed *p*-value (*p* = 0.00000002) for testing non-correlation. The grayscale filling depicts the comparison against the expert annotation scores. **b** Relationship between pixel-wise uncertainty and expert agreement (at least one expert with differing annotation; upper plot) and average prediction error rate (relative frequency of deviations between different expert segmentations and the predicted segmentation; lower plot) on the test set. Source data are provided as a Source Data file.

partly out-of-distribution images (same properties with previously unseen structures such as blood vessels), and 32 fully out-of-distribution images (different immunofluorescent labels) (Fig. 5b–d). Using the uncertainty score for sorting, the lowest uncertainty ranks are entirely taken by the 32 fully out-of-distribution images. Most of the partly out-of-distribution images obtain uncertainty ranks between 33 and 150 (Fig. 5a). A conservative protocol could require scientists to verify all images with an uncertainty score exceeding the reference uncertainty scores (Section "Quality Assurance"). Out-of-distribution images may then be excluded from the analysis or annotated for retraining in an active learning manner[21].

**Evaluation in the biomedical imaging wild**
So far, the evaluation of our study has been focused on ambiguous fluorescent images, as the underlying datasets allow us to demonstrate the use of deepflash2 along the entire bioimage analysis pipeline. However, deepflash2 can out-of-the-box deliver convincing segmentation results for other types of 2D images with an arbitrary number of input channels. Also, multiclass GT estimation, as well as multiclass semantic or instance segmentation, are supported. We showcase the use and performance of deepflash2 on three distinct biomedical imaging datasets that were part of recent data science challenges (Fig. 6, see Section "Evaluation metrics" for detailed dataset descriptions). We used default training parameter settings for all datasets except for the *gleason* dataset, where we adjusted a single hyperparameter to account for the large tumor regions (we increased the receptive field of the image tiles by selecting a zoom-out factor of 4).

The *gleason* challenge (2019) aims at the automatic Gleason grading (multiclass semantic segmentation) of prostate cancer from H&E-stained histopathology images[22]. The grading of prostate cancer tissue performed by different expert pathologists suffers from high inter-expert variability. deepflash2 outperforms the *nnunet* baseline Fig. 6 (last column) on all classes except the third class (very rare Gleason grade 5).

The *monuseg* (2018) challenge aims at nuclei segmentation in digital microscopic tissue images[23]. In this binary instance segmentation task, deepflash2 also outperforms the *nnunet* baseline and would

have reached a Top-10 rank in the challenge *monuseg* leaderboard yielding 0.67 in the challenge metric Aggrated Jaccard Index.

Finally, the recent *conic* (2022) challenge also aims at nuclei segmentation of H&E-stained histology images. The challenge is based on the Lizard dataset[24] containing half a million labeled nuclei in colon tissue and requires multiclass instance segmentation. deepflash2 outperforms the *nnunet* baseline Fig. 6 (last column) on all classes except the fourth class (Eosinophil).

## Discussion
The deepflash2 DL pipelines facilitate the objective and reliable segmentation of ambiguous bioimages integrating multi-expert annotations, deep model ensembles, and quality monitoring. They may thereby offer a blueprint for the training, evaluation, and application of DL in bioimaging projects, as they can be used with any tool or in custom DL pipelines.

As a tool, deepflash2 supports various use-cases for the integration of expert annotations, e.g., one annotation per image, multiple annotations per image (can be achieved by providing the same image under different names for each annotation), or training on the est. GT. Here, we want to discuss the best use of multi-expert annotations. These can help to mitigate the emerging DL replication crisis in the bioimage analysis as single-expert annotations may introduce errors or bias into model training[25]. Recall that image feature annotation is a complex perception task for humans and is subject to the individual annotator's graphical perceptual abilities[26]. Clear labeling instructions are of special importance to reduce the need for multi-expert annotations, as highlighted by Rädsch et al.[27].

There is not a per-se best annotation strategy, but the choice will rather depend on the bioimaging project and the available resources, i.e., we need to trade-off the number of training images, which should represent the diversity of the data, against the annotation quality gains from multiple annotations. Unbiased and precise annotations are typically acquired via GT estimation from multiple experts. Also, the repeated annotation of the same images allows us to approximate a human performance level on the given data, which is part of our proposed two-step evaluation process. Yet, repeated labeling of identical images results in a markedly higher annotation effort for each

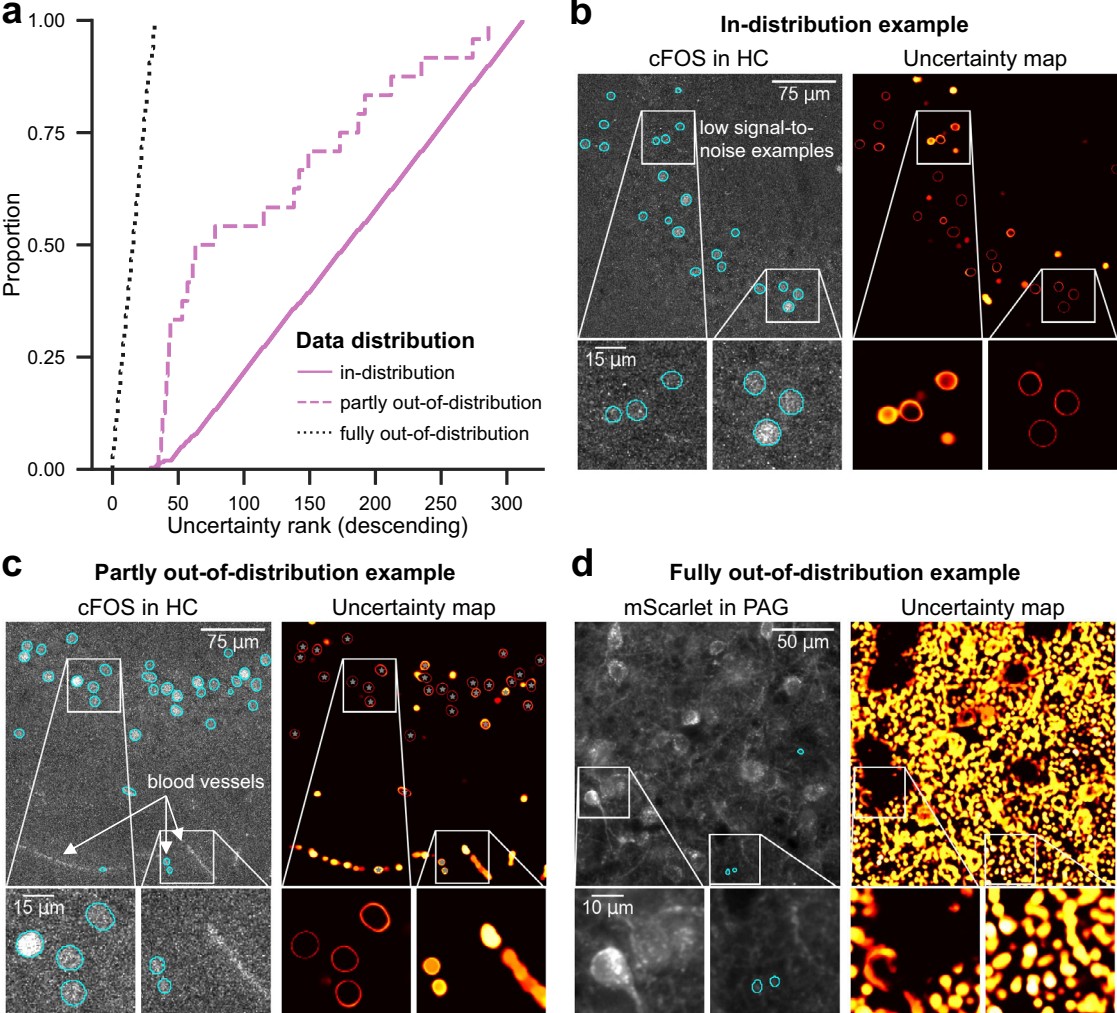

**Fig. 5 | Out-of-distribution detection. a** Out-of-distribution (ood) detection performance using heuristic ranking via uncertainty score. Starting the manual verification of the predictions at the lowest rank, all images with deviant fluorescence labels (fully ood, $N = 32$ images) are detected first. The partly ood images with previously unseen structures ($N = 24$) are mostly located in the lower ranks, and the in-distribution images (similar to training data of cFOS in HC, $N = 264$) are in the upper ranks. **b–d** Representative image crops of the three categories used in (**a**). Source data are provided as a Source Data file.

training image. Given a fixed annotation budget, multi-expert annotations would directly reduce the number of training images, which can have a detrimental effect on the predictive model performance if the underlying data distribution is not captured sufficiently. To obtain a better understanding of the annotation strategy trade-offs, we conducted some initial experiments regarding the most efficient use of expert time (see Supplementary Notes S4). We compared two strategies over different annotation budgets: The first strategy required the images to be annotated by all available experts. The second strategy required the experts to annotate different images, resulting in larger training sets. The results indicate that the second strategy is superior when only a few image annotations are available (small annotation budget). In this case, the model performance benefits from more (but less precise) image-annotation-pairs to capture the diverse data distribution. The first strategy is superior when more training annotations are available (higher annotation budget). Our results suggest that the consensus segmentations are indeed learnable by the DL models.

deepflash2 builds upon the integration of established DL libraries. For segmentation architectures such as the U-Net[3], deepflash2 leverages the *segmentation-models-pytorch* library[28]. The library has a large record of use in data science competition-winning solutions (see the "Hall of Fame"[28]), including deepflash2's Gold Medal and Innovation Award in the Kaggle data science competition hosted by the HuBMAP consortium[29]. Moreover, the encoder architectures of these segmentation models are based on the *timm* library[30], which has emerged as the de-facto benchmark DL library for image classification and is continuously updated with the latest model architectures, including the currently used ConvNext encoder[31]. There is currently no bioimaging tool making these resources easily accessible to life science researchers. Also, deepflash2's capability to automatically integrate new encoders and pretrained weights is a significant advantage over existing tools in the rapidly materializing field of DL.

By offering uncertainty measures (uncertainty maps, uncertainty score $U$), deepflash2 facilitates the aforementioned quality assurance procedure. Exploiting these measures in the bioimage analysis process promises insights into experimental conditions as well as biological mechanisms. Uncertainty arises from biological processes in experimental groups, for instance, when signal-to-noise ratios change due to global changes in image feature expression levels. Recognizing such quality issues during prediction offers a valuable feedback loop from analysis to experiment design and execution.

Initiatives such as the BioImage Model Zoo[32] or the Hugging Face Model Hub (https://huggingface.co/models) are simplifying DL model sharing in the research community. deepflash2 simplifies sharing of trained model ensembles, and we highly encourage scientists in making their research reproducible, accessible, and transparent. As

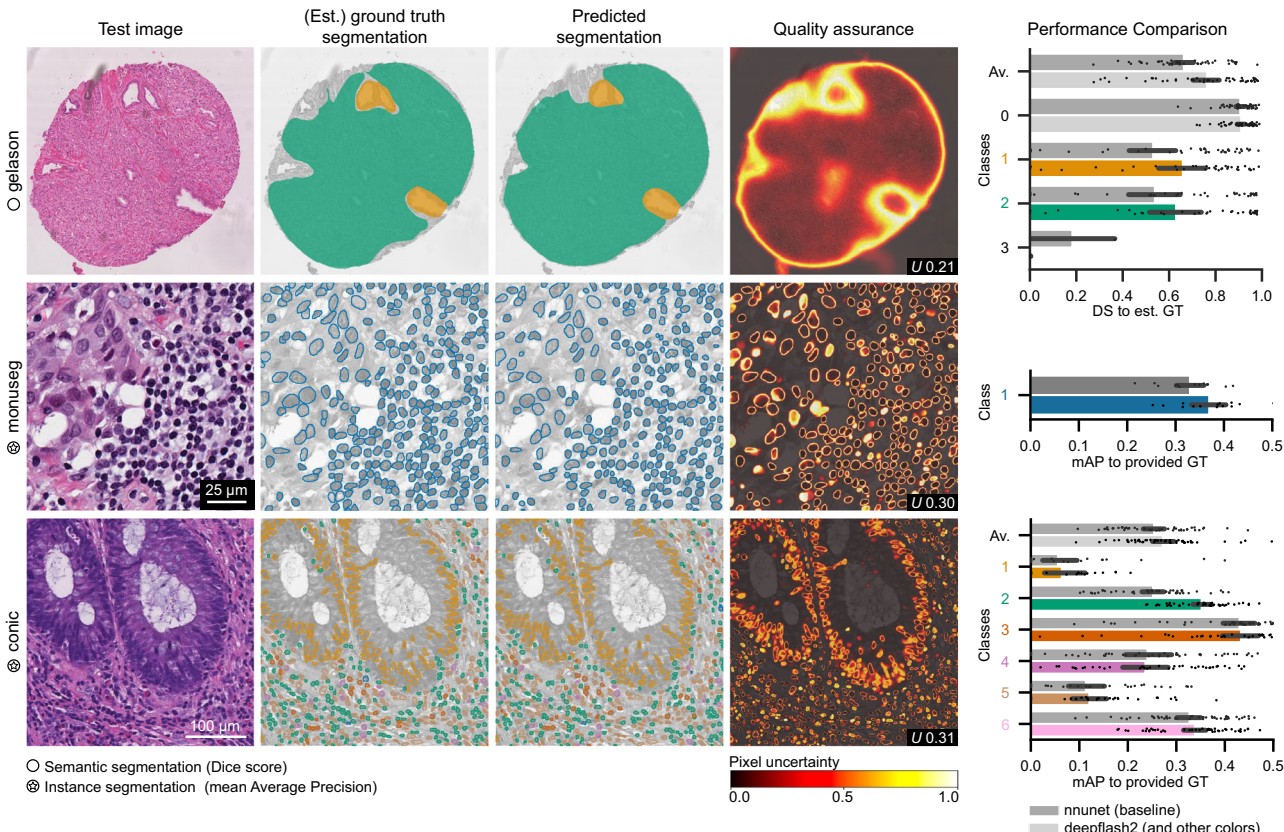

**Fig. 6 | Demonstration on challenge datasets *gleason, monuseg, conic*.** Exemplary test image slices (first column), corresponding GT segmentations (second column), predicted segmentations (third column), and uncertainty maps (fourth column) with uncertainty scores *U*. GT segmentations for the *gleason* dataset were estimated via STAPLE. The bar plots in the last column summarize the results over the entire test sets by class for semantic segmentation (*gleason*, N = 49 test images) and instance segmentation (*monuseg* N = 15 test images, *conic* N = 48 test images). The color codes in the *y*-axis labels and bars of the bar charts indicate the different class numbers in the segmentation masks (first and second row). We additionally report the average score across all classes (Av.) in multiclass settings. The error bars depict the 95% confidence interval of the observations estimated via bootstrapping around the arithmetic mean (center). Source data are provided as a Source Data file.

deepflash2 addresses the segmentation of ambiguous data that potentially varies across experiments, we think that a rigorous and transparent evaluation, as well as an easily accessible demonstration of the model's capabilities, can contribute to build trust in new, DL-enabled research.

deepflash2 aims to be a tool with preconfigured settings that offer out-of-the-box, very high predictive accuracy for *typical* bio-imaging tasks. However, this comes with some rigidity concerning the chosen hyperparameters. This may limit the tool's predictive performance on some datasets using default settings. A case in point was the *gleason* dataset, where we had to adjust the scaling factor to accommodate untypically large input images, which we could not capture with our default 512 × 512 patch sizes—such a manual expert adjustment of course runs against the goal of user-friendliness (note that *nnunet*, which automatically configures hyperparameters during training, does not face this problem). The proposed quality assurance procedure offers a direct assessment of the training data representativeness for a particular test instance by answering the question: How well-suited is the trained model ensemble for assessing this very instance? However, it does not provide any formal guarantees on the overall performance of the model ensemble and should be interpreted with caution. Ultimately, the reported uncertainty measures are influenced by the underlying DL models, training procedures, and the theoretical disentanglement between epistemic and aleatoric uncertainty (Section "Uncertainty quantification" and Supplementary Fig. S5.1).

deepflash2 offers an end-to-end integration of DL pipelines for bioimage analysis of ambiguous data. An easy-to-use GUI allows researchers without programming experience to rapidly train performant and robust DL model ensembles and monitor their predictions on new data. We are confident that deepflash2 can help establish more objectivity and reproducibility in natural sciences while lowering the overall workload for human annotators. deepflash2 introduces a concept for objective bioimage analysis that goes beyond ground truth estimation and measures of predictive accuracy. It also introduces ambiguity not only as a technical but also as a biological data variable in the bioimage analysis process. We think that this concept can serve as a baseline for DL-based biomedical image feature segmentation. Going forward, the tool will benefit from a growing user base which in turn helps reveal image specifications for which the default parameters may be less suitable. Subsequent releases will try to address such instances by establishing useful alternative configurations.

## Methods

### Ethical statement

All experiments and experimental procedures were in accordance with the guidelines set by the European Union and our local veterinary authority (Veterinäramt der Stadt Würzburg). In addition, all experiments and experimental procedures were approved by our institutional Animal Care, the Utilization Committee, and the Regierung von Unterfranken, Würzburg, Germany (License numbers: 55.2-2531.01-95/13 and 55.2.2-352-2-509/1067).

### Implementation details

The deepflash2 code library is implemented in Python 3, using *numpy*, *scipy*, and *opencv* for the base operations. The ground truth estimation

functionalities are based on the *simpleITK*[33]. The DL-related part is built upon the rich ecosystem of *PyTorch*[34] libraries, comprising *fastai*[35] for the training procedure, *segmentation models pytorch*[28] for segmentation architectures, *timm*[30] for pretrained encoders, and *albumentations*[36] for data augmentations. Instance segmentation capabilities are complemented using the *cellpose* library[9]. The trained model ensembles are designed to be directly executed in *ImageJ* using the *DeepImageJ* Plugin[37], can be shared on the BioImage Model Zoo[32], or hosted for inference. The deepflash2 GUI is based on the *Jupyter Notebook* environment[38]. Using interactive widgets[39] deepflash2 allows users to execute all analysis steps directly in the GUI or use the export functionality for subsequent processing in other tools (e.g., *ImageJ* or *Fiji*). Statistical analyses in this study were performed using *pingouin*; Figures were created using *seaborn* and *matplotlib*.

### Ground truth estimation

To train reproducible and unbiased models, deepflash2 relies on GT estimation from the annotations of multiple experts. deepflash2 offers GT estimation via simultaneous truth and performance level estimation (STAPLE)[14] (default in our analyses) or majority voting. Note that due to the ambiguities in the data, GT estimation can yield biologically implausible results (e.g., by merging the areas of two cells). We corrected such artifacts in our test sets. deepflash2 supports both multi-expert joining as well as single-expert annotations.

### Training procedure

The training of deepflash2 model ensembles is designed to achieve out-of-the-box rapid and high-quality segmentation of most bioimages without custom tuning. To achieve this, the deepflash2 pipeline was developed in an iterative manner seeking to establish a reliable base configuration.

The starting point for the selection parameter process was the award-winning solution at the Kaggle data science competition *HuBMAP - Hacking the Kidney* (see Section "Discussion"). To obtain a computationally manageable search space, we conducted some initial experiments on the training sets of the immunofluorescence data (*PV in HC, cFOS in HC, mScarlet in PAG, YFP in CTX*, and *GFAP in CTX*) via *k*-fold cross-validation. During this preselection phase, we fixed the architecture of our neural network as well as the weight initializations. Subsequently, we set up large-scale computational experiments to define the remaining hyperparameters via Bayesian optimization using *sweeps* on the Weights & Biases[40] MLOps platform. The search spaces included different encoders (ResNet18-50, EfficientNet b0-b4, ConvNext tiny and standard), tile shapes ($256 \times 256$, $512 \times 512$, $1024 \times 1024$), mini-batch sizes (2, 4, 8, 16, 32), learning rates (0.00001–0.01) for the Adam optimizer[41] with decoupled weight decay (0.00001–0.1), and training iterations (100–10,000). The *sweeps* were also evaluated on the immunofluorescence datasets. The training procedure for individual applications is outlined below.

### Default settings and customization options

The default DL-model architecture in deepflash2 is a U-net[3] with a ConvNext Tiny encoder[31]. The encoder is initialized with ImageNet[42] pretrained weights to allow better feature extraction and fast training convergence. The remaining weights in the segmentation architecture are initialized from a truncated normal distribution[43]. By combining pretraining and random initialization, this approach improves diversity in model ensembles. The encoder architectures were pretrained on 3-channel input images. If the new data has fewer than three input channels, we remove the excess pretrained weights in the first layer. If the new data comprises more than three input channels, we initialize the weights from a truncated normal distribution. Similar to the *nnunet*, we chose the mean of the cross-entropy and Dice loss[44] as the learning objective.

Each model is trained using the fine-tune policy of the *fastai* library[35]. This entails freezing the encoder weights, one-cycle training[45] of one epoch, unfreezing the weights, and again one-cycle training. During each epoch, we sample equally sized patches from each image in the training data. To address the issue of class imbalances, we use a weighted random sampling approach that ensures that the center points of the patches are sampled equally from each class. This kind of sampling also contributes to the data augmentation pipeline. Data augmentation operations include random augmentations such as rotating, flipping, and gamma correction; again, this follows best practices established by *nnunet*. We trained each model with one epoch in the first (frozen encoder weights) cycle and 25 epochs in the second cycle using a mini-batch size of four (patch size $512 \times 512$), a base learning rate of 0.001 and decoupled weight decay (0.001). We used a scale factor of 4 (zoom-out) for the *gleason* dataset and a scale factor of 1 for all other datasets (scaling is only applied during training and does not change the size of the final predictions). The training and validation data for the different models are shuffled by means of a *k*-fold cross-validation (with $k = 5$ in our experiments).

While they were designed for out-of-the-box usage, the deepflash2 Python API and GUI allow us to easily change all configuration parameters. These parameter choices can also be imported and exported via a JSON file. Experienced users can select alternative architectures (e.g., Unet++[46] or DeepLabV3+[47]) and encoders (e.g., ResNet[48], EfficientNet[49]). This flexibility is facilitated by the *segmentation models pytorch* package[28]. deepflash2 also provides options for common segmentation loss functions such as Focal[50], Tversky[51], or Lovasz[52]. Users can also adjust augmentation strategies or add more augmentations (e.g., contrast limited adaptive histogram equalization or grid distortions). One can also customize all training settings, for example, by opting for a different optimizer or setting a dataset-specific learning rate using the learning rate finder.

### Semantic segmentation

For the semantic segmentation of a new image with features $\mathbf{X} \in \mathbb{R}^{d \times c}$ deepflash2 predicts a semantic segmentation map $\mathbf{y} \in \{1, ..., K\}^d$, with $K$ being the number of classes, $d$ the dimensions of the input, and $c$ the input channels. Without loss of generality, class 1 is defined as background. We use the trained ensemble of $M$ deep neural networks to model the probabilistic predictive distribution $p_\theta(\mathbf{y}|\mathbf{X})$, where $\theta = (\theta_1, ..., \theta_M)$ are the parameters of the ensemble. Here, we leverage a sliding window approach with overlapping borders and Gaussian importance weighting[8]. We improve the prediction accuracy and robustness using $T$ deterministic test-time augmentations (rotating and flipping the input image). Each augmentation $t \in \{1, ..., T\}$ applied to an input image creates an augmented feature matrix $\mathbf{X_t}$. To combine all predictions, we follow Lakshminarayanan et al.[17] and treat the ensemble as a uniformly weighted mixture model to derive

$$p(\mathbf{y}|\mathbf{X}) = \frac{1}{T}\sum_{t=1}^{T}\frac{1}{M}\sum_{m=1}^{M} p_{\theta_m}(\mathbf{y}|\mathbf{X_t}, \theta_m) \qquad (1)$$

with $p_{\theta_m}(\mathbf{y}|\mathbf{X_t}, \theta_m) = \text{Softmax}(f_{\theta_m}(\mathbf{X_t}))$ and $f_{\theta_m}$ representing the neural network parametrized with $\theta_m$. We use $M = 5$ models and $T = 4$ augmentations in all our experiments. Finally, we obtain the predicted segmentation map

$$\hat{\mathbf{y}} = \underset{\mathbf{k} \in \{1, ..., K\}^d}{\text{argmax}} \; p(\mathbf{y} = \mathbf{k}|\mathbf{X}). \qquad (2)$$

### Uncertainty quantification

The uncertainty is typically categorized into aleatoric (statistical or per-measurement) uncertainty and epistemic (systematic or model) uncertainty[53]. To approximate the uncertainty maps of the predicted

segmentations, we follow the approach of Kwon et al.[54]. Here, we replace the Monte-Carlo dropout approach of Gal and Ghahramani[55] with deep ensembles, which have proven to produce well-calibrated uncertainty estimates and a more robust out-of-distribution detection[17]. In combination with test-time augmentations (inspired by Wang et al.[56]), we approximate the predictive (hybrid) uncertainty for each class $k \in \{1, \ldots, K\}$ as

$$
\text{Var}_{p(\mathbf{y}=k|\mathbf{X})} := \underbrace{\frac{1}{T}\sum_{t=1}^{T}\frac{1}{M}\sum_{m=1}^{M}\left[p_{\theta_m}(\mathbf{y}=k \mid \mathbf{X}_t, \theta_m) - p_{\theta_m}(\mathbf{y}=k \mid \mathbf{X}_t, \theta_m)^2\right]}_{\text{epistemic uncertainty}}
$$
$$
+ \underbrace{\frac{1}{T}\sum_{t=1}^{T}\frac{1}{M}\sum_{m=1}^{M}\left[p_{\theta_m}(\mathbf{y}=k \mid \mathbf{X}_t, \theta_m) - p(\mathbf{y}=k \mid \mathbf{X})\right]^2}_{\text{aleatoric uncertainty}}
$$

$$(3)$$

where $p(\mathbf{y}=k|\mathbf{X})$ denotes probabilities of a single class $k$.

To allow an intuitive visualization and efficient calculation in multiclass settings, we aggregate the results of the single classes to retrieve the final predictive uncertainty map:

$$
\text{Var}_{p(\mathbf{y}|\mathbf{X},\theta)} = \frac{\zeta}{K}\sum_{k=1}^{K}\text{Var}_{p(\mathbf{y}=k|\mathbf{X},\theta)} \tag{4}
$$

where $\zeta$ is a scaling factor. Following the derivation in Kwon et al.[54], the moment-based predictive uncertainty $\text{Var}_{p(\mathbf{y}=k|\mathbf{X})} \in [0; 0.25]$. Therefore, we set $\zeta$ to 4 in our experiments which scales the theoretical maximal pixel uncertainty to 1. Note that the formulation in Equation (4) may differ from the general formulation in Kwon et al.[54] for $K > 2$.

For the heuristic sorting and out-of-distribution detection, we define an aggregated uncertainty metric on image level. Let $\hat{y}_i$ be the predicted segmentation of pixel $i$, $\mathbf{x}_i$ the feature vector of pixel $i$ and $N$ the total number of pixels defined by $d$. We define the scalar-valued foreground uncertainty score for all predicted $N_f = \{i \in \{1, \ldots, N\}|\hat{y}_i > 1\}$ as

$$
U_{p(\mathbf{y}|\mathbf{X},\theta)} := \frac{1}{|N_f|}\sum_{i \in N_f}\text{Var}_{p(y_i|\mathbf{x}_i,\theta)}. \tag{5}
$$

**Instance segmentation**

If the segmented image contains touching objects (e.g., cells that are in close proximity), deepflash2 integrates the *cellpose* library[9], a generalist algorithm for cell and nucleus segmentation. We use the combined predictions of each class $p(\mathbf{y}=k|\mathbf{X})$ to predict the flow representations with the pretrained *cellpose* models. We then leverage the post-processing pipeline of *cellpose* to derive instance segmentations by combining the flow representations with the predicted segmentation maps $\hat{y}$. This procedure scales to an arbitrary number of classes and is, in contrast to the original *cellpose* implementation, not limited to one (or two) input channels. However, it requires the image feature shapes to be compatible with the pretrained *cellpose* models. To monitor the compatibility deepflash2 automatically reports the number of pixels that were removed during the instance segmentation process in the results table (column *cellpose_removed_pixels*). The differences were negligible in our experiments (<0.005%). We recommend increasing the *cellpose* flow threshold, which is directly adjustable in the deepflash2 GUI, or fine-tuning the *cellpose* models if these differences become more significant.

**Evaluation metrics**

For semantic segmentation, we calculate the similarity of two segmentation masks $\mathbf{y}_a$ and $\mathbf{y}_b$ using the Dice score. For binary masks, this metric is defined as

$$
\text{DS} := \frac{2\text{TP}}{2\text{TP} + \text{FP} + \text{FN}}, \tag{6}
$$

where the true positives (TP) are the sum of all matching positive (pixels) elements of $\mathbf{y}_a$ and $\mathbf{y}_b$, and the false positives (FP) and false negatives (FN) are the sum of positive elements that only appear in $\mathbf{y}_a$ or $\mathbf{y}_b$, respectively. In multiclass settings, we use macro averaging, i.e., we calculate the metrics for each class and then find their unweighted mean. The Dice score is commonly used for semantic segmentation tasks but is unaware of different instances (sets of pixels belonging to a class and instance).

For instance segmentation, let $\mathbf{y}_a^I$ and $\mathbf{y}_b^I$ be two instance segmentation masks that contain a finite number of instances $I_a$ and $I_b$, respectively. An instance $I_a$ is considered a match (true positive—$\text{TP}_\eta$) if an instance $I_b$ exists with an Intersection over Union (also known as Jaccard index) $\text{IoU}(I_a, I_b) = \frac{I_a \cap I_b}{I_a \cup I_b}$ exceeding a threshold $\eta \in (0,1]$. Unmatched instances $I_a$ are considered as false positives ($\text{FP}_\eta$), and unmatched instances $I_b$ as false negatives ($\text{FN}_\eta$). We define the Average Precision at a fixed threshold $\eta$ as $\text{AP}_\eta := \frac{\text{TP}_\eta}{\text{TP}_\eta + \text{FN}_\eta + \text{FP}_\eta}$. To become independent of fixed values for $\eta$, it is common to average the results over different $\eta$. The resulting metric is known as mean Average Precision and is defined as

$$
\text{mAP} := \frac{1}{|H|}\sum_{\eta \in H}\text{AP}_\eta. \tag{7}
$$

We use a set of 10 thresholds $H = \{\eta \in [0.50, \ldots, 0.95]|\eta \equiv 0 \bmod 0.05\}$ for all evaluations. This corresponds to the metric used in the COCO object detection challenge[57]. Additionally, we exclude all instances $I$ that are below a biologically viable size from the analysis. The minimum size is derived from the smallest area annotated by a human expert: 61 pixel (*PV in HC*), 30 pixel (*cFOS in HC*), 385 pixel (*mScarlet in PAG*), 193 pixel (*YFP in CTX*, 38 pixel (*monuseg*), and 3–6 pixel (*conic*).

**Quality assurance**

Once the deepflash2 model ensemble is deployed for predictions on new data, the quality assurance process helps the user prioritize the review of more ambiguous or out-of-distribution images. The predictions on such images are typically error-prone and exhibit a higher uncertainty score $U$. Thus, deepflash2 automatically sorts the predictions by decreasing the uncertainty score. Depending on the ambiguities in the data and the expected prediction quality (inferred from the hold-out test set), a conservative protocol could require scientists to verify all images with an uncertainty score exceeding a threshold $U_{min}$. Given the hold-out test set $Q = \{(\mathbf{X}_1, \mathbf{y}_1), \ldots, (\mathbf{X}_L, \mathbf{y}_L)\}$ where L is the number of samples, we define

$$
U_{min} := \min\left\{U_{p(\mathbf{y}|\mathbf{X},\theta)}|(\mathbf{y},\mathbf{X}) \in Q, S(y,\hat{y}) < \tau\right\} \tag{8}
$$

with $S(y,\hat{y})$ being an arbitrary evaluation metric (e.g., DS or mAP) and $\tau \in [0,1]$, a threshold that satisfies the prediction quality requirements. From a practical perspective, this means selecting all predictions from the test set with a score below the predefined threshold (e.g., DS = 0.8) and taking their minimum uncertainty score value $U$ as $U_{min}$. The verification process of a single image is simplified by the uncertainty maps that allow the user to quickly find difficult or ambiguous areas within the image.

**Evaluation datasets**

We evaluate our pipeline on five datasets that represent common bioimage analysis settings. The datasets exemplify a range of

fluorescently labeled (sub-)cellular targets in mouse brain tissue with varying degrees of data ambiguity.

The *PV in HC* dataset published by Segebarth et al.[11] describes indirect immunofluorescence labeling of Parvalbumin-positive (PV-positive) interneurons in the hippocampus. Morphological features are widely ramified axons projecting to neighbored neurons for soma-near inhibition of excitatory neuronal activity[58]. The axonal projections densely wrap around the somata of target cells. This occasionally causes data ambiguities when the somata of the PV-positive neurons need to be separated from the PV-positive immunofluorescent signal around the soma of neighbored cells. Thresholding approaches such as Otsu's method (see Supplementary Note S2.2) typically fail at this task as it requires differentiating between rather brightly labeled somata that express PV in the cytosol vs. brightly labeled PV-positive axon bundles that can appear in the neighborhood.

The publicly available *cFOS in HC* dataset[59] describes indirect immunofluorescent labeling of the transcription factor cFOS in different subregions of the hippocampus after behavioral testing of the mice[11]. The counting or segmentation of cFOS-positive nuclei is an often-used experimental paradigm in the neurosciences. The staining is used to investigate information processing in neural circuits[60]. The low SNR of cFOS labels for most but not all image features renders its heuristic segmentation a very challenging task. This results in a very high inter-expert variability after manual segmentation (see Segebarth et al.[11] and Supplementary Fig. S2.1). We use 280 additional images of this dataset to demonstrate the out-of-distribution detection capabilities of deepflash2. There are no expert annotations available for the additional images; however, 24 images comprise characteristics that do not occur in the training data. We classified such partly out-of-distribution images into three different error categories for our study: blood vessels if the images contained blood vessels (13 images); folded tissue (4 images); fluorescent particles if there was at least one strongly fluorescent particle unrelated to the actual fluorescent label (7 images) (see examples in Supplementary Fig. S1.1).

The *mScarlet in the PAG* dataset shows an indirect immunofluorescent post-labeling of the red-fluorescent protein mScarlet, after viral expression in the periaqueductal gray (PAG). Here, microscopy images visualize mScarlet, tagged to the light-sensitive inhibitory opsin OPN3. The recombinant protein was delivered via stereotactic injection of an adeno-associated viral vector (AAV2/5-Ef1a-DIO-eOPN3-ts-mScarlet-ER) to the PAG. Optogenetics is a key technology in neuroscience that allows the control of neuronal activity in selected neuronal populations[61,62]. Consequently, the number of opsin-expressing neurons provides highly relevant information in optogenetic experiments. However, due to the substantial efforts that these analyses require, this data is rarely acquired[2]. Therefore, we chose this dataset of a recombinant opsin that shows a particularly low signal-to-noise ratio (Fig. 2) in order to evaluate the usability of deepflash2 for this commonly requested use-case.

The *YFP in CTX* dataset shows direct fluorescence of yellow fluorescent protein (YFP) in the cortex of so-called thy1-YFP mice. In thy1-YFP mice, a fluorescent protein is expressed in the cytosol of neuronal subtypes with the help of promoter elements from the thy1 gene[63]. This provides a fluorescent Golgi-like vital stain that can be used to investigate disease-related changes in neuron numbers or neuron morphology, for instance, for hypothesis-generating research in neurodegenerative diseases (e.g., Alzheimer's disease). Here, computational bioimage analysis is aggravated by the pure intensity of the label that causes strong background signals by light scattering or out-of-focus light. Both can blur the signal borders in the image plane.

Finally, the *GFAP in HC* dataset shows indirect immunofluorescence signals of glial acidic fibrillary protein (GFAP) in the hippocampus. Anti-GFAP labeling is one of the most commonly used stainings in the neurosciences and is also used for histological examination of brain tumor tissue. Glial cells labeled by GFAP in the hippocampus show different morphologies (e.g., radial-like or star-like). GFAP-positive cells occupy separate anatomical parts[64] (like balls in a ball bath). Thus, it is highly laborious to manually segment the spatial area of GFAP-positive single astrocytes in a brain slice. Here, the extensions of the GFAP-labeled astrocytic skeleton cannot be separated from parts of neighboring astrocytes, rendering a reliable instance separation and thus instance segmentation impossible. Albeit the signal is typically bright and very clear around the center of the cell, the signal borders of the radial fibers become ambiguous due to the 3D-ball-like structure, low SNR at the end of the fibers, and out-of-focus light interference.

A high-level comparison of the key dataset characteristics is provided in Table 1.

### Challenge datasets

We additionally evaluate the performance of deepflash2 on three recent biomedical imaging challenge datasets. The *gleason* challenge (2019) aims at the automatic Gleason grading (multiclass semantic segmentation) of prostate cancer from H&E-stained histopathology images[22]. The grading of prostate cancer tissue performed by different expert pathologists suffers from high inter-expert variability. Ground truth estimation was performed using STAPLE[14]. For undecided pixels, we assigned the segmentation of the expert with the highest score. Class 0 corresponds to benign or other tissue, class 1 to Gleason grade 3, class 2 to Gleason grade 4, and class 3 to Gleason grade 5.

The *monuseg* (2018) challenge aims at nuclei segmentation in digital microscopic tissue images[23]. The task is binary instance segmentation (class 1 nucleus, class 0 other/background).

The *conic* (2022) challenge aims at nuclei segmentation of H&E-stained histology images. The challenge is based on the Lizard dataset[24]. The images were acquired with a 20x objective magnification (about 0.5 microns/pixel) from six different data sources. They contain half a million labeled nuclei in colon tissue and require multiclass

**Table 1 | Comparison of immunofluorescence datasets**

|  | PV in HC | cFOS in HC | mScarlet in PAG | YFP in CTX | GFAP in HC |
|---|---|---|---|---|---|
| Annotation target | Somata | Nuclei | Somata | Somata | Morphology |
| Semantic segmentation | Yes | Yes | Yes | Yes | Yes |
| Instance segmentation | Yes | Yes | Yes | Yes | No |
| Train images | 36 | 36 | 12 | 12 | 12 |
| Test images | 8 | 8 | 8 | 8 | 8 |
| Experts | 5 | 5 | 4–5 | 4–5 | 3 |
| Additional images | – | 280 | – | – | – |
| Fluorescence microsc. | Confocal | Confocal | Light | Light | Light |
| Size (pixel) | 1024×1024 | 1024×1024 | 2752×2208 | 2752×2208 | 580×580 |
| Resolution (px/μm) | 1.61 | 1.61 | 3.7 | 3.7 | 3.7 |

instance segmentation. Here, class 1 corresponds to the category epithelial, class 2 to lymphocyte, class 3 to plasma, class 4 to eosinophil, class 5 to neutrophil, and class 6 to connective tissue.

## Performance benchmarks

We benchmark the predictive performance of deepflash2 against a select group of well-established algorithms and tools. These comprise the U-Net[2] and *nnunet*[8] for both semantic and instance segmentation as well as two out-of-the-box baselines. We utilize Otsu's method[18] as a simple baseline for semantic segmentation and *cellpose*[9] as a generic baseline for (cell) instance segmentation. Additionally, we benchmark deepflash2 against fine-tuned *cellpose* models and ensembles, showing superior performance of our method (see Supplementary Table S2.1). *cellpose* has previously proven to outperform other well-known methods for instance segmentation (e.g., Mask-RCNN[19] or StarDist[20]).

For each dataset, we apply the tools as described by their developers to render the comparison as fair as possible. We train the U-Net[2] on a 90/10 train-validation-split for 10,000 iterations (learning rate of 0.00001 and the Adam optimizer[41]) using the authors' *TensorFlow 1.x* implementation. This includes all relevant features, such as overlapping tile strategy and border-aware loss function. We derive the parameter values for the loss function (border weight factor ($\lambda$), border weight sigma ($\sigma_{sep}$), and foreground-background ratio ($v_{bal}$) by means of Bayesian hyperparameter tuning: *Parv in HC*: $\lambda = 25$, $\sigma_{sep} = 10$, $v_{bal} = 0.66$; *cFOS in HC*: $\lambda = 44$, $\sigma_{sep} = 2$, $v_{bal} = 0.23$; *mScarlet in PAG*: $\lambda = 15$, $\sigma_{sep} = 10$, $v_{bal} = 0.66$; *YFP in CTX*: $\lambda = 15$, $\sigma_{sep} = 5$, $v_{bal} = 0.85$; *GFAP in HC*: $\lambda = 1$, $\sigma_{sep} = 1$, $v_{bal} = 0.85$.

We train the self-configuring *nnunet* (version 1.6.6) model ensemble[8] following the authors' instructions provided on GitHub.

*cellpose* provides three pretrained model ensembles (*nuclei*, *cyto*, and *cyto2*) for out-of-the-box usage[9]. We select the ensemble with the highest score on the training data: *cyto* for Parv in HC and YFP in CTX, *cyto2* for cFOS in HC, and mScarlet in PAG. During inference, we fix the cell diameter (in pixel) for each dataset: Parv in HC: 24; cFOS in HC: 15; mScarlet in PAG: 55; YFP in CTX: 50. We additionally provide a performance comparison for fine-tuned *cellpose* models and ensembles in Supplementary Note S2.2. We use the *cellpose* GitHub version with commit hash 316927e (August 26, 2021) for our experiments.

We repeat our experiments with different seeds to ensure that our results are robust and reproducible (see Supplementary Note S2.2). The experiments for training duration comparison are executed on the free platform Google Colaboratory (Nvidia Tesla K80 GPU, 2 vCPUs; times were extrapolated when the 12-h limit was reached) and the paid Google Cloud Platform (Nvidia A100 GPU, 12 vCPUs). The remaining experiments are executed locally (Nvidia GeForce RTX 3090) or in the cloud (Google Cloud Platform on Nvidia Tesla K40 GPUs).

## Experimental animals

The datasets *mScarlet in PAG*, *YFP in CTX*, and *GFAP in HC* were acquired for this study. Here, all mice were bred in the animal facility of the Institute of Clinical Neurobiology at the University Hospital of Würzburg, Germany, and housed under standard conditions ($55 \pm 5\%$ humidity, $21 \pm 1$ C, 12:12-h light:dark cycle) with access to food and water ad libitum. VGlut2-IRES-Cre knock-in mice[65] (stock no. 208863), as well as Thy1-YFP mice[63] (stock no. 003782), were obtained from Jackson Laboratory. Additionally, we used wild-type mice with the genetic background C57BL/6J (Charles River, CRL:027). Only male mice at ages between 4 and 8 months were used.

## Surgeries

The surgeries for mice in *mScarlet in PAG* were conducted as follows: Male *VGlut2-IRES-Cre* knock-in mice were injected at the age of 4 months, and adeno-associated virus (AAV) was used as vectors to deliver genetic material into the brain. AAV vectors encoding Cre-dependently for the inhibitory opsin eOPN3[66] were injected into the periaqueductal gray (PAG) bilaterally. The construct *EF1α-DIO-eOPN3-ts-mScarlet-ER* was kindly provided by Simon Wiegert, Center for Molecular Neurobiology Hamburg, Germany. Respective AAV vectors were produced in house (AAV2/5 capsid). For stereotactic surgeries, animals were prepared with an administration of Buprenorphin (Buprenorvet, Bayer). Mice were deeply anesthetized with 4–5% isoflurane/O2 (Anesthetic Vaporizer, Harvard Apparatus). Animals were fixed into the stereotactic frame (Kopf, Model 1900), and anesthesia was maintained with 1.5–2% isoflurane/O2. Subcutaneous injection of Ropivacaine (Naropin Aspen) was used for local analgesia before opening the scalp. Craniotomies were performed at bregma coordinates AP −4.5 mm, ML ±0.6 mm. A glass pipette (Drummond Scientific) was filled with the viral vector and lowered to the target depth of −2.9 mm from bregma. A volume of 100 nl was injected with a pressure injector (NPI electronic). After injection, the pipette was held in place for 8 min before retracting. The wound was closed, and the animal was treated with a subcutaneous injection of Metacam (Metacam, Boehringer Ingelheim) for post-surgery analgesia. After 6 weeks of expression time, animals were perfused, and brain tissue was dissected for further analysis.

## Sample preparation

Following intraperitoneal injection (for *YFP in CTX* and *GFAP in HC*): 12 µl/g bodyweight of a mixture of ketamine (100 mg/kg; Ursotamin, Serumwerk) and xylazine (16 mg/kg; cp-Pharma, Xylavet, Burgdorf, Germany); for *mScarlet in PAG*: urethane (2g/kg; Sigma-Aldrich) at a volume of 200 µl diluted in 0.9% sterile sodium chloride solution), the depth of the anesthesia was assessed for each mouse by testing the tail and the hind limb pedal reflexes. Upon absence of both reflexes, mice were transcardially perfused using phosphate-buffered saline (PBS) with (for *YFP in CTX* and *GFAP in HC*) or without (*mScarlet in PAG*) 0.4% heparin (Heparin-Natrium-25000, ratiopharm), and subsequently a 4% paraformaldehyde solution in PBS for fixation. After dissection, brains were kept in 4% paraformaldehyde solution in PBS for another 2 h (for *YFP in CTX* and *GFAP in HC*) or overnight (for *mScarlet in PAG*) at 4 °C. Brains were then washed twice with PBS and stored at 4 °C until sectioning. For cutting, brains were embedded in 6% agarose in PBS, and a vibratome (Leica VT1200) was used to cut 40 µm (for *YFP in CTX* and *GFAP in HC*) or 60 µm (for *mScarlet in PAG*) coronal sections. Immunohistochemistry was performed in 24-well plates with up to three free-floating sections per well in 400 µl solution and under constant shaking.

For *YFP in CTX* and *GFAP in HC*: brain sections were incubated for 1 h at room temperature in 100 mM Tris-buffered glycine solution (pH 7.4). Slices were then incubated with blocking solution (10% horse serum, 0.3% Triton X100, 0.1% Tween 20, in PBS) for 1 h at room temperature. Subsequently, sections were labeled with primary antibodies at the indicated dilutions in blocking solution for 48 h at 4 °C (rabbit anti-GFAP, Acris, DP014, 1:200; chicken anti-GFP, Abcam, Ab13970, 1:1000). Primary antibody solutions were washed off thrice with washing solution (0.1% Triton X100 and 0.1% Tween 20 solution in PBS) for 10 min each. Sections were then incubated with fluorescently labeled secondary antibodies at 0.5 µg/ml in blocking solution for 1.5 h at room temperature (goat anti-chicken Alexa-488 conjugated, Invitrogen; donkey anti-rabbit Cy3 conjugated, Jackson ImmunoResearch). Finally, sections were incubated again twice for 10 min with the washing solution and once with PBS at room temperature, prior to embedding in Aqua-Poly/Mount (Polysciences).

For *mScarlet in PAG*: brain sections were incubated in blocking solution (10% donkey serum, 0.3% Triton X100, 0.1% Tween in 1x TBS) for 2 h at room temperature. For labeling, sections were incubated for 2 days at 4 °C with rabbit anti-RFP (Biomol, 600-401-379, 1:1000) in 10% blocking solution in 1X TBS-T. Sections were washed thrice with washing solution for 10 min each and then incubated with the fluorescently labeled secondary antibody at 0.5 µg/ml (donkey anti-rabbit

Cy3, Jackson ImmunoResearch). Following a single wash with 1x TBS-T for 20 min at room temperature, sections were incubated with DAPI (Roth, 6335.1, 1:5000) in TBS-T for 5 min and eventually washed twice with 1x TBS-T. The labeled sections were embedded in an embedding medium (2.4 g Mowiol, 6 g Glycerol, 6 ml ddH2O, diluted in 12 ml 0.2 M Tris at pH 8.5).

## Image acquisition, processing, and manual analysis

Image acquisition for *mScarlet in PAG*, *YFP in CTX*, and *GFAP in HC* was performed using a Zeiss Axio Zoom.V16 microscope, equipped with a Zeiss HXP 200C light source, an Axiocam 506 mono camera, and an APO Z 1.5x/0.37 FWD 30 mm objective. Images covering 743.7 × 596.7 μm of the corresponding brain regions at a resolution of 3.7 px/μm were acquired as 8-bit images. To foster manual ROI annotation, these raw 8-bit images were enhanced for brightness and contrast using the automatic brightness and contrast enhancer implemented in Fiji[67]. The corresponding image features of interest were manually annotated by Ph.D.-level neuroscientists.

## Statistics and reproducibility

To evaluate the predictive performance of deepflash2 and the benchmark tools, we used the train-test split from Segebarth et al.[11] for the *PV in HC* and *cFOS in HC* datasets. Here, we removed one image (id 1608) from each test set to ensure a balanced evaluation with eight test images across all five fluorescent datasets in this study. The datasets *mScarlet in PAG*, *YFP in CTX*, and *GFAP in HC* were randomly split into 12 images used for training and eight images used for evaluation. The challenge datasets were randomly split into 80% train and 20% test data, resulting in 196 training and 49 test images for the *gleason* dataset and 190 training and 48 test images for the *conic* dataset. For the *monuseg* dataset, we used the provided train-test split from the challenge, comprising 30 training and 15 test images.

All computational experiments were independently repeated three times with similar results.

No statistical method was used to predetermine the sample size. The experts were not blinded during the image annotation process; however, they did not receive information on the annotations of the other experts.

## Reporting summary

Further information on research design is available in the Nature Portfolio Reporting Summary linked to this article.

# Data availability

The data and trained DL models generated in this study have been deposited on Zenodo[68]. The train and test data for *cFOS in HC* can also be downloaded from Dryad[59]. The external challenge data is available at the challenge websites for *gleason*[23,69], *monuseg*[22,70], and *conic*[24,71]. Source data are provided with this paper.

# Code availability

The source code is publicly available on GitHub[72]. The repository also contains Jupyter notebooks with instructions to easily reproduce the paper's analyses and benchmark methods on Google Colab. Additionally, the documentation[73] provides walk-through tutorials and videos for using the GUI as well as information on the deepflash2 Python API.

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

## Acknowledgements

We thank Toni Greif and Kai Günder for critically reviewing the mathematical content. We thank Annemarie Schulte for her valuable deepflash2 user feedback. We thank Friederike Griebel for the design of the deepflash2 logo. The research of R.B. was funded by the Deutsche Forschungsgemeinschaft (DFG, German Research Foundation) project-ID 424778381-TRR 295 and 426503586-KFO5001. The research of P.T. is supported by the Deutsche Forschungsgemeinschaft through Heisenberg professorship and project funds (TO 1124/1,2,3), TRR 295 (424778381), and a NARSAD Young Investigator Grant of the Brain and Behavior Foundation. This publication was supported by the Open Access Publication Fund of the University of Wuerzburg.

## Author contributions

M.G., D.S., N.St., R.B., and C.M.F. conceptualized this study. M.G. designed and implemented the deepflash2 Python API and GUI, wrote the documentation, implemented testing and continuous integration, executed the computational experiments, and prepared all figures. M.G., D.S., N.St., and C.M.F. selected and designed the computational experiments. M.G., N.St., and C.M.F. formalized the uncertainties. D.S., N.Sc., R.B., and P.T. created the neurobiological datasets and did the animal experimentation. D.S., N.Sc., and R.B. did the brain slice IHC and annotated the bioimages. D.S. and N.Sc. performed confocal/light microscopy to generate the image data. M.G., D.S., and N.St. wrote the original manuscript. R.B. and C.M.F. reviewed and edited the manuscript. N.Sc. and P.T. reviewed and contributed to the improvement of the manuscript.

## Funding

## Competing interests

The authors declare no competing interests.
