## [Peer Review File · Nature Communications]

Reviewers' Comments:

Reviewer #1:

Remarks to the Author:

The authors have addressed all three of my major comments with new analyses, which I appreciate. Of these three, I want to follow up on one of the comments, the one about most efficient use of annotator time. A considerable dataset of annotations seems to have been generated for this new analysis (which I recommend to be shared publicly). I found it confusing to determine how many images each were labelled for 1,2 etc experts in the two strategies, and as far as I understand the strategies are the same for one expert. I am still not sure I understand how many different images were annotated by all five experts for each dataset. My first suggestion is to show the comparisons only for the 5 expert datasets, since that should provide the clearest differences.

My second suggestion is to show the comparisons as a function of the number of training images, rather than as a function of the number of experts. Annotating 35 images is a relatively high annotation investment, and the performance of the models probably saturates considerably before that. Thus, a clearer picture might emerge if the performance is plotted vs # training images.

I don't really have any other comments. As always, please share in the final version the annotated data collected for this paper.

Reviewer #2:

Remarks to the Author:

My major concerns of limited accessibility to non-computer scientists was nicely resolved by adding more extensive tutorials and the option/perspective to execute the trained models in established tools like ImageJ and the option to host those models on the BioImage Model Zoo. The remaining comments were also answered/resolved properly and there are no further changes requested from my side.

Reviewer #3:

Remarks to the Author:

I thank the authors for their very detailed response. The paper gained a lot since the first iteration and large parts of my feedback are now incorporated. Particularly the participation in competitions with deepflash2 and its good performance there provides compelling evidence for its value proposition! This is complemented by a much improved documentation of the software which makes it substantially more user-friendly than it initially was.

Nonetheless there are still some points that I would consider unresolved and that I think warrant further discussions, experiments or changes to the manuscript. I will do my best to sort them thematically so that we can focus our discussion.

Deepflash2 development

As with any machine learning method, it is imperative to distinguish the data a method is developed on from the data it is evaluated on. The paper still does not provide sufficient details on how the hyperparameters used in deepflash2 (training scheme, sampling, loss, learning rate, data augmentation etc etc) were identified by the authors and what datasets were used during this

process. All information I could gather in that regard was that the encoder was selected based on a 5-fold cross-validation on the immunofluorescence data. This is something that should be clarified.

Deepflash2 application

Nowhere in the paper is clearly stated how deepflash2 was applied to the datasets that were used for evaluation. With the promise of being an out-of-the-box tool, I would expect that no hyperparameters were manually adjusted between datasets. Otherwise there would not be much of a point to the method, because adjusting hyperparameters is a task that requires expertise that the targeted user-base might not have. For the gleason dataset, the authors seem to have added a 4x downscaling to the data!?

If deepflash2 was indeed applied out of the box without any manual changes to its configuration (which I believe is mostly the case) then this is certainly information that the authors will want to place as prominently as possible in the paper.

Deepflash2's feature set

The way things are currently communicated, an out-of-the-box applicable default setting seems somewhat contradictory with the plethora of encoders, features and hyperparameters that deepflash offers at the same time. Some straightening of the communication might be needed in the sense that the defaults are perfectly fine for end-users and that these additional features are not needed for a good user experience. Highlighting that these target more experienced users only and understanding them is not required for using the method might be a good strategy.

Training from scratch

I do not agree with the assessment of the authors that fine tuning from an ImageNet pretrained model can be called 'training from scratch'. There are datasets for which ImageNet fine tuning works better than others, so it is essential to understand how reliant deepflash2 is on this pretraining to provide added value. Thus, experiments showing true training from scratch (all weights randomly initialized) are needed. This would also ensure fair comparison with nnU-Net which intentionally does not support imagenet pretrained encoders in order to be more flexible in its configuration (including its ability to process 3d data which deepflash does not support at all). On a side note: How does deepflash handle different numbers on input channels when imagenet pretrained models expect 3? This could be problematic if more than 3 channels are available.

More thorough discussion on deepflash's limitations

Even though deepflash2 has again shown strong performance, its rigid set of hyperparameters (as opposed to nnU-Net which automatically configures some of them) might cause it to fail on some (yet unseen) datasets where nnU-Net might succeed. This limitation might need more discussion, including potential scenarios where deepflash2 might not perform well (for instance a segmentation problem where the target structures cannot be recognized within the 512x512 patch size that is always used).

Use of annotation budget

I appreciate the addition of experiments regarding the best use of annotation budget. However, as the authors state themselves, the experiments are not conclusive yet, or at least only show a small benefit in a "low SNR" environment (based on just one dataset). Given these preliminary results I find the author's maintained stance questionable that multiple experts per image are absolutely necessary. ("To train reproducible and unbiased models deepflash2 relies on GT estimation from the annotations of multiple experts." Line 281). I would highly appreciate it if the authors would either extend their experiments to find conclusive evidence for their claims or just rephrase that multi-expert joining is supported just like single expert annotations.

Where I am quite confident that the 'multiple annotation per case is better than annotating more cases' hypothesis falls apart is when the data distribution is more diverse (i.e. the appearance of the target structures varies a lot, there are rare imaging artifacts etc). In such a case it would be detrimental to favor more precise annotations over presenting more samples which can lead to increased robustness of the model. While such a scenario does not seem to be presented in the datasets that were used for evaluation here, such situations are likely to appear in the wild.

Consequently I would like to see some effort to explore this, for example via an extension of the experiments where the "budget" for the annotations (currently all samples are used when all 5

experts are used) is reduced to fewer and fewer samples, corresponding to a smaller budget. This should elucidate potential (dis)advantages of either approach even if the data distribution is more or less homogenous.

On a tangent: the need for multi-experts could probably be reduced when giving clear labeling instructions, as highlighted in the paper Labeling instructions matter in biomedical image analysis <https://arxiv.org/abs/2207.09899>

On the use of multiple annotations per case

The paper the authors refer to in their response highlights three variants on how multiple labels per annotator could be used: 1) single models, each trained on one of the annotators 2) a model trained on the consensus annotations and 3) an ensemble of models trained on the consensus annotations. The variant I requested in my initial review, which in my view is the most intuitive approach to this problem, is (as far as I can tell) not investigated in that paper and also not discussed here: an ensemble of single models, each trained on one of the annotators (similar to 1) except that the expert models are not evaluated individually but used as an ensemble). If the authors are able to provide convincing evidence that the approach used in deepflash2 outperforms this baseline then I am more than happy to concede this point.

Rather than insisting on the use of multiple labels per case, the authors could simply highlight that deepflash2 supports any desired use-case (one label per case, multiple labels per case) and let the user decide. In this case some guidance should be added about what strategy is useful in different scenarios.

Identifying and using uncertainties/ambiguities

Given the importance the authors attribute to the extraction and utilization of uncertainty in the manuscript there are several aspects that seem lacking:

The computation of uncertainties uses a combination of monte carlo dropout and ensembling, where all models in the ensemble were trained using consensus annotations. I think the authors are wasting potential here: training an ensemble of expert models (each trained on the annotation of one expert) could yield much more informative uncertainty estimates. If this (more intuitive) approach is not selected, there should be experiments demonstrating its inferiority

The response to R3-7 is a bit dissatisfactory. Especially when only some experts found a certain object (while others labeled it as background) this ambiguity is completely lost as soon as a consensus segmentation is formed. The fact that qualitatively some of these cases seem to be captured by the uncertainty estimation is not proof that these cases can be captured by deepflash2. Training models on non-hard labels or an ensemble of expert models should be able to provide more robust uncertainty estimates. Again, additional experiments could underline the author's claims

A bit of quantification could be useful, for example: Given a certain uncertainty estimation scheme (ensemble of consensus (+ monte carlo) vs ensemble of expert models (+ monte carlo)), how many images need to be evaluated to capture x% of the failure cases?

Measuring image-level uncertainty by (if I understood correctly) averaging the model uncertainty over the pixels belonging to a certain class can cause biases towards images with many small predicted objects (large object boundary relative to object center), like the one shown in Fig 5 d. Have alternative aggregation methods been explored?

Minor points:

Table 1 should be extended to include all datasets (or separate table for non-immunofluorescence datasets)

Line 341: "Note that, due to symmetries, the the ..."

Line 373: The threshold should probably not include 0 but be higher than 0, as ≥ 0 would mean a guaranteed True positive for all instances even if dice = 0.

Summary: The paper has received a lot of attention and has clearly improved in many aspects. There are, however, still some areas where more information and experimental validation is needed (see above). Particularly, details on how the hyperparameter configuration for deepflash was performed (and using which datasets), how it was applied to the remaining datasets and what manual changes, if any, needed to be made, are not described sufficiently. In addition, the design choice of using consensus annotations still seems odd, lacking sufficient and convincing experimental evidence over the more intuitive approach of using an ensemble of models each

trained on the annotations of one expert. This also touches on the uncertainty estimation part which, from my perspective, could really benefit from an explicit use of inter-rater variability. Finally, the requirement to annotate each sample multiple times over simply annotating more samples seems to be the less robust choice when considering that datasets 'in the wild' can present highly varying morphologies and imaging artifacts.

Dear Reviewers,

We again thank you for your critical assessment of our study “*Deep learning in the bioimaging wild: Handling ambiguous data with deepflash2.*” We appreciate the helpful insights, hints, and ideas and incorporated them into the revised manuscript. In the following, we address the remarks of the review team point by point (for each reviewer individually). Please find our changes in the document with the tracked changes.

Sincerely,

Matthias Griebel, Nikolai Stein, and Christoph M. Flath

Reviewer 1

R1 Summary — *The authors have addressed all three of my major comments with new analyses, which I appreciate. Of these three, I want to follow up on one of the comments, the one about most efficient use of annotator time.*

R1 - 1 — *A considerable dataset of annotations seems to have been generated for this new analysis (which I recommend to be shared publicly). I found it confusing to determine how many images each were labeled for 1,2 etc experts in the two strategies, and as far as I understand the strategies are the same for one expert. I am still not sure I understand how many different images were annotated by all five experts for each dataset.*

Reply: Thank you for bringing up this point. Leveraging the immunofluorescence datasets that are already used throughout our study (and are already publicly available) we conducted computational experiments for this analysis: We repeatedly simulated the different annotation strategies by sampling from the pool of available training images and expert annotations. We have revised and clarified the description of this procedure in this revision.

R1 - 2 — *My first suggestion is to show the comparisons only for the 5 expert datasets, since that should provide the clearest differences.*

Reply: Thank you for this suggestion! The datasets *PV in HC* and *cFOS in HC* are not only annotated by five experts but also contain the highest number of training images (36), which is favorable for this kind of analysis. Our revised plots (see Fig. S4.1) now only display these two datasets and provide a clearer picture of the different annotation strategies: the DIFFERENT strategy is advantageous when only a few annotations are available, while the STAPLE strategy becomes favorable when more annotations are available (higher annotation effort). The other immunofluorescence datasets were simply too small and thus more confusing than helpful: with only 12 training images, the simulation of the STAPLE strategy would end with only four images for training on the est. ground truth from three experts.

R1 - 3 — *My second suggestion is to show the comparisons as a function of the number of training images, rather than as a function of the number of experts. Annotating 35 images is a relatively high*

annotation investment, and the performance of the models probably saturates considerably before that. Thus, a clearer picture might emerge if the performance is plotted vs # training images.

Reply: This suggestion was very helpful; thank you! We have conducted several new computational experiments with different numbers of training annotations for this revision. Showing the performance plotted vs. the number of training annotations indeed shows a clearer picture (see explanation above).

Reviewer 2

R 2 Summary — *My major concerns of limited accessibility to non-computer scientists was nicely resolved by adding more extensive tutorials and the option/perspective to execute the trained models in established tools like ImageJ and the option to host those models on the BioImage Model Zoo. The remaining comments were also answered/resolved properly, and there are no further changes requested from my side.*

Reply: We are pleased that we could address and resolve your concerns appropriately. Thank you again for your helpful insights and ideas.

Reviewer 3

R3 Summary — *I thank the authors for their very detailed response. The paper gained a lot since the first iteration and large parts of my feedback are now incorporated. Particularly the participation in competitions with deepflash2 and its good performance there provides compelling evidence for its value proposition! This is complemented by a much improved documentation of the software which makes it substantially more user-friendly than it initially was. Nonetheless there are still some points that I would consider unresolved and that I think warrant further discussions, experiments or changes to the manuscript. I will do my best to sort them thematically so that we can focus our discussion.*

Reply: We greatly appreciated the constructive and insightful comments on the manuscript. They helped us both with respect to formulations in the text as well as complementary analyses strengthening our arguments.

R3-1 — ***Deepflash2 development:*** *As with any machine learning method, it is imperative to distinguish the data a method is developed on from the data it is evaluated on. The paper still does not provide sufficient details on how the hyperparameters used in deepflash2 (training scheme, sampling, loss, learning rate, data augmentation etc etc) were identified by the authors and what datasets were used during this process. All information I could gather in that regard was that the encoder was selected based on a 5-fold cross-validation on the immunofluorescence data. This is something that should be clarified.*

Reply: Thank you for raising this concern. We agree that we have not provided sufficient details on how the hyperparameters used in deepflash2 were identified. We have added an entire paragraph concerning the development and hyperparameter selection of deepflash2 in the Methods Section of this revision.

R3-2 — *Deepflash2 application:* Nowhere in the paper is clearly stated how deepflash2 was applied to the datasets that were used for evaluation. With the promise of being an out-of-the-box tool, I would expect that no hyperparameters were manually adjusted between datasets. Otherwise there would not be much of a point to the method, because adjusting hyperparameters is a task that requires expertise that the targeted user-base might not have. For the gleason dataset, the authors seem to have added a 4x downscaling to the data!? If deepflash2 was indeed applied out of the box without any manual changes to its configuration (which I believe is mostly the case) then this is certainly information that the authors will want to place as prominently as possible in the paper.

Reply: deepflash2 was indeed applied “in the wild” without any changes except for a single parameter for the gleason dataset. We added a more detailed explanation why we increased the receptive field of the image tiles (zoom-out factor of 4) to account for the large tumor regions.

R3-3 — *Deepflash2’s feature set:* The way things are currently communicated, an out-of-the-box applicable default setting seems somewhat contradictory with the plethora of encoders, features and hyperparameters that deepflash offers at the same time. Some straightening of the communication might be needed in the sense that the defaults are perfectly fine for end-users and that these additional features are not needed for a good user experience. Highlighting that these target more experienced users only and understanding them is not required for using the method might be a good strategy.

Reply: We rewrote and restructured the corresponding part in the methods section (4.2). The reviewer comment was particularly helpful for us to highlight deepflash2’s hybrid focus on ease of use and accuracy.

R3-4 — *Training from scratch:* I do not agree with the assessment of the authors that fine tuning from an ImageNet pretrained model can be called ‘training from scratch’. There are datasets for which ImageNet fine tuning works better than others, so it is essential to understand how reliant deepflash2 is on this pretraining to provide added value. Thus, experiments showing true training from scratch (all weights randomly initialized) are needed. This would also ensure fair comparison with nnU-Net which intentionally does not support imagenet pretrained encoders in order to be more flexible in its configuration (including its ability to process 3d data which deepflash does not support at all). On a side note: How does deepflash handle different numbers on input channels when imagenet pretrained models expect 3? This could be problematic if more than 3 channels are available.

Reply: Thank you for raising this issue; we agree that using a pre-trained encoder is not “training from scratch”, and we have removed such claims from the paper. We have further conducted experiments on all immunofluorescence datasets with training from scratch (all weights randomly initialized) and included the results in Table S2.1. The results highlight the importance of pre-training: Without a pre-trained encoder, deepflash2 is mostly outperformed by the nnU-Net.

R3-5 — *On a side note: How does deepflash handle different numbers on input channels when imagenet pretrained models expect 3? This could be problematic if more than 3 channels are available.*

Reply: We have added the implementation details regarding the input channels in the Methods section: If the new data has ≤ 3 input channels, we (partly) reuse the pre-trained weights in the first layer. If the new data comprises > 3 input channels we initialize the weights from a truncated normal distribution.

R3-6 — *More thorough discussion on deepflash’s limitations: Even though deepflash2 has again shown strong performance, its rigid set of hyperparameters (as opposed to nnU-Net which automatically configures some of them) might cause it to fail on some (yet unseen) datasets where nnU-Net might succeed. This limitation might need more discussion, including potential scenarios where deepflash2 might not perform well (for instance a segmentation problem where the target structures cannot be recognized within the 512x512 patch size that is always used).*

Reply: We explicated the tradeoff stemming from rigid parameter choices (speed and usability versus flexibility) in the discussion with a dedicated limitations paragraph. We agree that the rigid set of hyperparameters might be a Achille’s heel of *deepflash2* for datasets with special properties. In such cases, a more flexible architecture (e.g., nnU-Net) may succeed. We connect this discussion with your remark on the parameters adjustment for the *gleason* data set where the *deepflash2* increases significantly by only changing one hyperparameter.

R3-7 — *Use of annotation budget: I appreciate the addition of experiments regarding the best use of annotation budget. However, as the authors state themselves, the experiments are not conclusive yet, or at least only show a small benefit in a “low SNR” environment (based on just one dataset). Given these preliminary results I find the author’s maintained stance questionable that multiple experts per image are absolutely necessary. (“To train reproducible and unbiased models deepflash2 relies on GT estimation from the annotations of multiple experts.” Line 281). I would highly appreciate it if the authors would either extend their experiments to find conclusive evidence for their claims or just rephrase that multi-expert joining is supported just like single expert annotations.*

Reply: Thank you for bringing this up! We have rephrased the sections accordingly.

R3-8 — *Where I am quite confident that the ‘multiple annotation per case is better than annotating more cases’ hypothesis falls apart is when the data distribution is more diverse (i.e. the appearance of the target structures varies a lot, there are rare imaging artifacts etc). In such a case it would be detrimental to favor more precise annotations over presenting more samples which can lead to increased robustness of the model. While such a scenario does not seem to be presented in the datasets that were used for evaluation here, such situations are likely to appear in the wild. Consequently I would like to see some effort to explore this, for example via an extension of the experiments where the “budget” for the annotations (currently all samples are used when all 5 experts are used) is reduced to fewer and fewer samples, corresponding to a smaller budget. This should elucidate potential (dis)advantages of either approach even if the data distribution is more or less homogenous.*

Reply: This was a really valuable suggestion which helped us frame our evaluation strategy in a more nuanced manner. We have extended our experiments with varying annotation budgets (Fig. S.4.1), which was also suggested by Reviewer 1. Our results show that it is indeed favorable to use a different annotation strategy for small annotation budgets which confirms your chain of arguments. We have adjusted the Discussion section accordingly.

R3-9 — *On a tangent: the need for multi-experts could probably be reduced when giving clear labeling instructions, as highlighted in the paper Labeling instructions matter in biomedical image analysis <https://arxiv.org/abs/2207.09899>*

Reply: Thank you for this insightful resource; we have integrated these comments and added a citation to the paper.

R3-10 — *On the use of multiple annotations per case: The paper the authors refer to in their response highlights three variants on how multiple labels per annotator could be used: 1) single models, each trained on one of the annotators 2) a model trained on the consensus annotations and 3) an ensemble of models trained on the consensus annotations. The variant I requested in my initial review, which in my view is the most intuitive approach to this problem, is (as far as I can tell) not investigated in that paper and also not discussed here: an ensemble of single models, each trained on one of the annotators (similar to 1) except that the expert models are not evaluated individually but used as an ensemble). If the authors are able to provide convincing evidence that the approach used in deepflash2 outperforms this baseline then I am more than happy to concede this point.*

Reply: We have conducted further experiments and found convincing evidence that the approach used in deepflash2 is generally as good or superior (see Appendix S3, Figure S3.1).

R3-11 — *Rather than insisting on the use of multiple labels per case, the authors could simply highlight that deepflash2 supports any desired use-case (one label per case, multiple labels per case) and let the user decide. In this case some guidance should be added about what strategy is useful in different scenarios.*

Reply: We adopted this alternative positioning and now highlight that deepflash2 supports either approach to labeling.

R3-12 — *Identifying and using uncertainties/ambiguities: Given the importance the authors attribute to the extraction and utilization of uncertainty in the manuscript there are several aspects that seem lacking: The computation of uncertainties uses a combination of monte carlo dropout and ensembling, where all models in the ensemble were trained using consensus annotations.*

Reply: We fear that our presentation of the uncertainty estimation set a wrong trail. Our current approach for uncertainty estimation is indeed based on Gal & Ghahramani (2016) but does not use Monte-Carlo dropout. Instead, we leverage model ensembles and test-time-augmentation.

R3-13 — *More intuitive approach: I think the authors are wasting potential here: training an ensemble of expert models (each trained on the annotation of one expert) could yield much more informative uncertainty estimates. If this (more intuitive) approach is not selected, there should be experiments demonstrating its inferiority.*

Reply: Very helpful to ask this question as we had tried the intuitive approach at an earlier point. The revision includes Supplement S3 which shows that the alternative approach yields inferior results.

R3-14 — *Especially when only some experts found a certain object (while others labeled it as background) this ambiguity is completely lost as soon as a consensus segmentation is formed. The*

fact that qualitatively some of these cases seem to be captured by the uncertainty estimation is not proof that these cases can be captured by deepflash2. Training models on non-hard labels or an ensemble of expert models should be able to provide more robust uncertainty estimates. Again, additional experiments could underline the author’s claims

Reply: We agree that it is unsatisfactory that the approach does not use all the available information but training on the est. GT delivered the best predictive performance. We deemed accuracy somewhat more important than better-calibrated uncertainties.

R3-15 — *Uncertainty quantification examples: A bit of quantification could be useful, for example: Given a certain uncertainty estimation scheme (ensemble of consensus (+ monte carlo) vs ensemble of expert models (+ monte carlo)), how many images need to be evaluated to capture x% of the failure cases?*

Reply: This is actually a nice idea but it seems contingent on uncertainty quantification using Monte Carlo dropout which we actually do not use (see R3-12). Consequently, pursuing this approach would be less straightforward and arguably somewhat confusing as it requires introducing a new kind of uncertainty estimation. For this reason we retained the current exposition.

R3-16 — *Image-level uncertainty from pixel averages: Measuring image-level uncertainty by (if I understood correctly) averaging the model uncertainty over the pixels belonging to a certain class can cause biases towards images with many small predicted objects (large object boundary relative to object center), like the one shown in Fig 5 d. Have alternative aggregation methods been explored?*

Reply: We have indeed explored various methods for measuring image-level uncertainty. For instance, we have trained One-Class SVMs for out-of-distribution detection on different feature sets derived from the uncertainty map (mean, median, variance, etc.). Furthermore, we tested energy-based approaches for uncertainty estimation based on Liu, Weitang, et al. “Energy-based out-of-distribution detection.” (Advances in Neural Information Processing Systems 33 (2020): 21464-21475.). However, the presented approach delivered the most reliable results.

R3 — Minor Points

- *Table 1 should be extended to include all datasets (or separate table for non-immunofluorescence datasets)*

Reply: We have added a table for the challenge datasets (Table S2.2)

- *Line 341: “Note that, due to symmetries, the the ...”*

Reply: Corrected, thank you!

- *Line 373: The threshold should probably not include 0 but be higher than 0, as $\zeta=0$ would mean a guaranteed True positive for all instances even if dice = 0.*

Reply: We agree that a threshold of 0 would make no sense at all. To exclude 0, we have replaced the square bracket with a parenthesis.

Reviewers' Comments:

Reviewer #1:

Remarks to the Author:

All good, thanks for making the changes.

Reviewer #3:

Remarks to the Author:

I thank the authors for addressing a plethora of the points I previously criticized with their revised manuscript. The paper has progressed a lot and the current version feels much more complete than the previous, especially the added experiment with the annotation budget is very interesting and can provide the practitioner with a good guideline when to invest time in annotating different examples rather than the same example multiple times.

For easier comprehension I will reiterate which points of criticism I raised before and how they were addressed:

Deepflash2 development (Status: Addressed)

Originally the process leading to the final hyperparameters was not clarified sufficiently, despite it being one of the most important parts of Deepflash2. With the added paragraph in Section 4.2 the authors highlight how they reached their current hyperparameters clarifying this process sufficiently.

Deepflash2's feature set (Status: Addressed)

Previously a clear communication that the many adjustments deepflash2 provides the user with are aimed at the advanced users and that good out-of-the-box performance is achievable with default parameters was missing. By adding the customization paragraph in Section 4.2, which details the process of discovering the default hyperparameters, this point is resolved.

Training from scratch (Status: Addressed)

Originally the authors talked about Deepflash2 being trained from scratch despite the encoder being pre trained on Imagenet. Additionally they did not provide ablations how much benefit this pre training brings, as compared to training from scratch. In the revision the authors removed the claims of training from scratch and also added an ablation of the effects of the pretraining in the Supplement which I liked a lot and shows that the pretraining is very important for the high out-of-the-box performance.

Additionally information on changes of the input channels are adapted for the pretrained models were missing. The authors addressed this question by stating that higher channel numbers get randomly initialized however how the model behaves when fewer channels are present is not entirely clear to me. "Partly recycling" could mean many different things, however I deem this a minor point of criticism, which could easily get explained more clearly.

More thorough discussion on deepflash's limitations (Status: Addressed)

Originally the limitations of Deepflash2 were not explored and no information on when Deepflash2 might fail were given. The authors added a Limitations section which highlights this issue for the gleason dataset where the authors had to intervene manually to improve the performance.

I would have liked seeing some more experiments probing where the default Deepflash2 setting breaks down (e.g. through more datasets). However, since the scope of Deepflash2 is communicated clearly and since finding many more public datasets for the domain Deepflash2 is designed for can be difficult I deem the point as sufficiently addressed.

Use of annotation budget (Status: Addressed)

In the previous manuscript the authors did not sufficiently motivate why 5 experts should label the same data points instead of labeling different data points. In the revised version they included an additional experiment that investigates the benefits of the different strategies, which provides practitioners with a guideline which labeling strategy might be best employed for their problem at hand. I am very happy to see that the authors incorporated our previous points of criticism by including this experiment.

Despite this, the current experiment is limited in scope and could be extended further e.g. by investigating some other dimensions which might influence which labeling strategy performs better (e.g. by checking datasets which are more/less ambiguous).

Identifying and using uncertainties/ambiguities (Status: Partially addressed)

In the previous manuscript version the way the authors deal with the uncertainties that are present through the aggregation of ground truths was not well substantiated and lacked comparisons to other methods of utilizing the data ambiguity.

In the revised version the authors included some of our criticisms about using the uncertainties by adding the baseline of an ensemble of expert models which were trained on the annotation of single experts. They show that this gets outperformed by the ensemble trained on STAPLE which motivates their current choice (S3).

Despite this a major point of criticism remains in the usefulness of the ambiguities for the user: In order to get value the uncertainty quantification has to show some benefit for the user. So far there is only the relative ranking that is provided by the authors which shows that iid samples are usually lower uncertainty than ood samples.

A hypothetical example on how many images a user would need to review and check to guarantee a segmentation quality greater than some DICE threshold is not provided which could be done easily and help quantify the benefits of the uncertainty estimation.

Overall I still believe the uncertainty ranking section needs to be reworked and show that this is actually useful, since this section is a large part of the whole manuscript and so far only provides qualitative evidence. Adding one experiment that quantifies how many examples would have to be reviewed to assure a certain segmentation quality (when ordering by uncertainty and with GT info) would be an easy way to show that the uncertainties actually bring value.

Summary:

The revision of the paper addressed many open points of criticism that I originally had. To name some of them, the authors:

...addressed the lack of thorough explanation how they arrived at their current hyperparameter settings.

... highlighted that deepflash is used with default hyperparameters and the limitations that it sometimes needs manual adjustment of the user.

... explicitly state now that deepflash2 can be used out-of-the-box but also features various adjustment possibilities a knowledgeable user can.

... added a paragraph on the limitations of deepflash2.

... provide an additional experiment which investigates the debated point of use of annotation time.

The only remaining point of criticism from my perspective remains that the authors still do not provide a quantitative measure of the potential benefit of their uncertainty measure. To me the current qualitative examples do not suffice to claim a feature to be beneficial. As I mentioned previously an experiment on how many samples would need to be reviewed to guarantee a certain annotation quality would not be very laborious and could provide some much needed quantification and motivation for the uncertainty feature of deepflash2.

Overall I am very happy with the progress of the paper and really appreciate the great effort the authors had to put in to address almost all points of criticism that the other Reviewers and I raised. The quality of the paper has already increased significantly and is up to a high standard, with only a few open points of criticism remaining.

Dear Reviewers,

We again thank you for your critical assessment of our study “*Deep learning in the bioimaging wild: Handling ambiguous data with deepflash2.*” We appreciate the helpful insights, hints, and ideas and incorporated them into the revised manuscript. In the following, we address the remarks – primarily of Reviewer 3 – point by point. Please find our changes in the document with the tracked changes.

Sincerely,

Matthias Griebel, Nikolai Stein, and Christoph M. Flath

Reviewer 1

R1 Summary — *All good, thanks for making the changes.*

Reply: We are pleased that we could address and resolve your concerns appropriately. Thank you again for your helpful insights and ideas.

Reviewer 2

n.a.

Reviewer 3

R3 Summary — *I thank the authors for addressing a plethora of the points I previously criticized with their revised manuscript. The paper has progressed a lot and the current version feels much more complete than the previous, especially the added experiment with the annotation budget is very interesting and can provide the practitioner with a good guideline when to invest time in annotating different examples rather than the same example multiple times.*

Reply: Thank you for your thorough review of our paper. We greatly appreciated your constructive and insightful comments on the manuscript. They helped us both with respect to the manuscript text as well as coming up with complementary analyses strengthening our arguments.

R3-1 — *For easier comprehension I will reiterate which points of criticism I raised before and how they were addressed:*

Deepflash2 development (Status: Addressed): *Originally the process leading to the final hyperparameters was not clarified sufficiently, despite it being one of the most important parts of Deepflash2. With the added paragraph in Section 4.2 the authors highlight how they reached their current hyperparameters clarifying this process sufficiently.*

Deepflash2’s feature set (Status: Addressed): *Previously a clear communication that the many adjustments deepflash2 provides the user with are aimed at the advanced users and that good out-of-the-box performance is achievable with default parameters was missing. By adding*

the customization paragraph in Section 4.2, which details the process of discovering the default hyperparameters, this point is resolved.

Training from scratch (Status: Addressed): Originally the authors talked about Deepflash2 being trained from scratch despite the encoder being pre trained on Imagenet. Additionally they did not provide ablations how much benefit this pre training brings, as compared to training from scratch. In the revision the authors removed the claims of training from scratch and also added an ablation of the effects of the pretraining in the Supplement which I liked a lot and shows that the pretraining is very important for the high out-of-the-box performance. Additionally information on changes of the input channels are adapted for the pretrained models were missing. The authors addressed this question by stating that higher channel numbers get randomly initialized however how the model behaves when fewer channels are present is not entirely clear to me. “Partly recycling” could mean many different things, however I deem this a minor point of criticism, which could easily get explained more clearly.

More thorough discussion on deepflash’s limitations (Status: Addressed): Originally the limitations of Deepflash2 were not explored and no information on when Deepflash2 might fail were given. The authors added a Limitations section which highlights this issue for the gleason dataset where the authors had to intervene manually to improve the performance. I would have liked seeing some more experiments probing where the default Deepflash2 setting breaks down (e.g. through more datasets). However, since the scope of Deepflash2 is communicated clearly and since finding many more public datasets for the domain Deepflash2 is designed for can be difficult I deem the point as sufficiently addressed.

Use of annotation budget (Status: Addressed): In the previous manuscript the authors did not sufficiently motivate why 5 experts should label the same data points instead of labeling different data points. In the revised version they included an additional experiment that investigates the benefits of the different strategies, which provides practitioners with a guideline which labeling strategy might be best employed for their problem at hand. I am very happy to see that the authors incorporated our previous points of criticism by including this experiment. Despite this, the current experiment is limited in scope and could be extended further e.g. by investigating some other dimensions which might influence which labeling strategy performs better (e.g. by checking datasets which are more/less ambiguous).

Reply: We are pleased that we were able to address these points. We felt that the manuscript and the analysis package have greatly benefited from these suggestions and edits. With this in mind, we also addressed your minor point of criticism and rephrased the sentence with the ambiguous term “Partly recycling” accordingly.

R3-2 — Identifying and using uncertainties/ambiguities (Status: Partially addressed): In the previous manuscript version the way the authors deal with the uncertainties that are present through the aggregation of ground truths was not well substantiated and lacked comparisons to other methods of utilizing the data ambiguity. In the revised version the authors included some of our criticisms about using the uncertainties by adding the baseline of an ensemble of expert models which were trained on the annotation of single experts. They show that this gets outperformed by the ensemble trained on STAPLE which motivates their current choice (S3).

Reply: We are pleased that the additional experiments in S3 provide compelling evidence to support our current modeling choices.

R3-3 — Identifying and using uncertainties/ambiguities (Status: Partially addressed):

Despite this a major point of criticism remains in the usefulness of the ambiguities for the user: In order to get value the uncertainty quantification has to show some benefit for the user. So far there is only the relative ranking that is provided by the authors which shows that iid samples are usually lower uncertainty than ood samples. A hypothetical example on how many images a user would need to review and check to guarantee a segmentation quality greater than some DICE threshold is not provided which could be done easily and help quantify the benefits of the uncertainty estimation. Overall I still believe the uncertainty ranking section needs to be reworked and show that this is actually useful, since this section is a large part of the whole manuscript and so far only provides qualitative evidence. Adding one experiment that quantifies how many examples would have to be reviewed to assure a certain segmentation quality (when ordering by uncertainty and with GT info) would be an easy way to show that the uncertainties actually bring value.

Reply: Thank you for another detailed review and for your constructive feedback. However, unlike with the other points raised we have a strong feeling that following this route would fundamentally alter the scope / character of the manuscript. We fully understand that it would be ideal to quantify the benefits of the uncertainty estimation in terms of the number of images that a user would need to review and check to achieve a segmentation quality above a certain DICE threshold. However, this would ultimately boil down to some kind of performance guarantee which is typically not in the scope of applied ML papers (with limited data material) but instead a core topic in ML theory (with a focus on asymptotic behavior for ever-growing data sets). Note that the uncertainty measure in our manuscript quantifies the classifier confidence for a given sample vis-à-vis a trained model (ensemble). For us and other research groups this has proven to be a helpful tool for answering a typical question: How well-suited is the trained model ensemble for assessing this very instance? This use case is highlighted and illustrated using the sorting procedure in Figure 5 in the manuscript. We updated the discussion of the uncertainty measures and the general limitations in Section 3 accordingly. In particular, we tried to avoid formulations suggesting a global quantification.

Note that the proposed (hypothetical) experiment implicitly assumes that suitable data is readily available. However, as noted in the last revision, our quantification relies on multiple annotations which are not readily obtained for larger data sets. Therefore, preparing and conducting such experiments would actually be quite laborious and add limited benefit as another entry in the supplement. For these reasons we have not adopted such an experiment.

R3-4 — Summary: *Overall I am very happy with the progress of the paper and really appreciate the great effort the authors had to put in to address almost all points of criticism that the other Reviewers and I raised. The quality of the paper has already increased significantly and is up to a high standard, with only a few open points of criticism remaining.*

Reply: We appreciate the fundamentally positive assessment and are hopeful that the non-inclusion of another experimental study does not impact the maturity level the manuscript has reached over the course of the review process.

Reviewers' Comments:

Reviewer #1:

Remarks to the Author:

I am reviewer R1, and I have evaluated the remaining concerns of reviewer R3. I agree with the authors that these concerns are beyond the scope of this manuscript, and the proposed additional work would take a huge investment to do well for relatively little extra information.

Dear Reviewers,

We again thank you for your critical assessment of our study “*Deep learning-enabled segmentation of ambiguous bioimages with deepflash2.*” We appreciate the helpful insights, hints, and ideas and incorporated them into the revised manuscript. In the following, we address the final remarks of Reviewer 1 point by point.

Sincerely,

Matthias Griebel, Nikolai Stein, and Christoph M. Flath

Reviewer 1

R1 Summary — *I am reviewer R1, and I have evaluated the remaining concerns of reviewer R3. I agree with the authors that these concerns are beyond the scope of this manuscript, and the proposed additional work would take a huge investment to do well for relatively little extra information.*

Reply: We are pleased that you agree with our assessment. Thank you again for all the effort you put into reviewing our work.